



# HYD-RESPONSES: daily hydro-meteorological catchment-level time series to analyse HYDrological drought dynamics in RESPONSE to (cumulative) water deficits in Swiss catchments.

Christoph Nathanael von Matt[1,2], Benjamin David Stocker[1,2], and Olivia Martius[1,2]

[1]Institute of Geography, University of Bern, Bern, Switzerland
[2]Oeschger Center for Climate Change Research, University of Bern, Bern, Switzerland

**Correspondence:** Christoph Nathanael von Matt (christoph.vonmatt@unibe.ch)

**Abstract.**

The HYD-RESPONSES dataset (https://doi.org/10.5281/zenodo.14713274; von Matt et al., 2025) provides new daily catchment-level time series for key hydro-meteorological variables necessary to study drought conditions, including precipitation, snow water equivalent, temperature, soil moisture, (potential) evaporation, and streamflow. The dataset covers 184 small to large Swiss catchments of the surface water monitoring network operated by the Federal Office for the Environment (FOEN). The catchments range across a variety of streamflow regime types, mean altitudes, biogeographic regions, and anthropogenic influences. The data set provides daily average streamflow derived from measurements by the FOEN and daily hydrometeorological data (precipitation, temperature, radiation, snow and soil moisture) on the catchment level extracted from spatially gridded data provided by MeteoSwiss (RhiresD, TabsD, TmaxD, TminD, SrelD), MeteoSwiss and the WSL Institute for Snow and Avalanche Research SLF (SPASS), SLF (OSHD), and the European Centre for Medium-Range Weather Forecasts ECMWF (ERA5-Land).

In addition, derived indicators describing snowfall, snowmelt, (potential) water balance and streamflow are provided. Information on precipitation, evaporation-driven and streamflow deficits are provided in form of standardized and non-standardized (drought/deficit) indices. Standardized indices include the SPI, SPEI and SMRI and are provided on multiple aggregation scales from 1 to 24 months (mostly in 3-monthly steps). Non-standardized indices are provided as cumulative (water) deficits in (potential) water balance (CWD and PCWD) and streamflow (CQD). For all variables and indices, the climatology and the (standardized) anomalies are available on various time scales (daily, monthly, seasonal, and yearly). Drought event time series containing drought event numbers and drought event durations, are provided for streamflow droughts identified by using two percentile-based event definitions (fixed and variable threshold) and for cumulative water deficits (CWD, PCWD and CQD). Detailed catchment descriptors covering hydro-climatological and hydro-terrestrial aspects as well as streamflow characteristics are provided for all catchments. The dataset can be used to study weather-driven streamflow extremes, to train data-driven machine-learning algorithms, to study drought propagation, and for comparative analyses of catchment responses in disturbed and undisturbed catchments. The dataset is compatible with the recently published CAMELS-CH dataset and with additional catchment descriptors provided by the FOEN.



## 1 Introduction

In recent years, the frequency of droughts has increased in Europe and Switzerland with notable drought years in 2003, 2011, 2015, 2018, 2020. Most recently, in 2022, conditions were characterized as unprecedented in terms of compound heat and drought in the last 500 years over large parts of Europe (BAFU, 2016; BAFU et al. (Hrsg.), 2019; BUWAL, BWG, MeteoSchweiz, 2004; Scherrer et al., 2022; Tripathy and Mishra, 2023). Under climate change, this trend is likely to continue with projected increases in drought frequency, dry spell duration, and drought severity for both individual and combined drought types (Brunner et al., 2019b, a; Calanca, 2007; Kotlarski et al., 2023; Muelchi et al., 2021a; von Matt et al., 2024). Increasing drought impacts on various sectors are expected. This has prompted Swiss national authorities to establish a national drought early warning system (DEWS, see https://www.trockenheit.admin.ch/en; BAFU (Hrsg.), 2021; CH2018, 2018; Haile et al., 2020; Henne et al., 2018; Naumann et al., 2021; Brunner et al., 2019a; Otero et al., 2023; Ranasinghe et al., 2021; Tschurr et al., 2020; BAFU, 2022; Swiss Confederation, 2025).

Droughts are an inherently multivariate phenomenon with often non-linear drought propagation from meteorological conditions to impacts on ecosystems, infrastructure, and economy. Individual drought events may differ in their hydro-climatological, hydro-meteorological, hydro-terrestrial and anthropogenic characteristics (Brunner et al., 2023; Hao and Singh, 2015; Mishra and Singh, 2010; Zhou et al., 2021; Floriancic et al., 2020; Massari et al., 2022). The consideration of multiple hydro-climatic, hydro-meteorological, hydro-terrestrial and anthropogenic factors is therefore key to understand catchment-specific drought responses and sensitivities and to provide information for drought early warning, preparations, and interventions (e.g., Apurv et al., 2017; Apurv and Cai, 2020; Baez-Villanueva et al., 2024; Brunner et al., 2022, 2021; Ding et al., 2021; Peña-Angulo et al., 2022; Peña-Gallardo et al., 2019; Sutanto and Van Lanen, 2022; Tijdeman et al., 2018; Van Lanen et al., 2013; Savelli et al., 2022; Van Loon and Laaha, 2015; von Matt et al., 2024).

Novel high-resolution observational datasets provide a unique opportunity to combine multiple hydro-meteorological variables to analyze and monitor drought dynamics and the evolution of drought impacts of individual events at the catchment-level. For example, the propagation of meteorological to hydrological droughts or the evolution of droughts from the development to the recovery phase can be studied (Brunner et al., 2021; Brunner and Chartier-Rescan, 2024; Parry et al., 2016; Raposo et al., 2023; Brocca et al., 2024; Brunner et al., 2021; Stocker et al., 2023; Poussin et al., 2021). The Federal Office for Climatology and Meteorology (MeteoSwiss) provides a suite of high-resolution essential climate variables spatially interpolated to a regular grid from a dense measurement station network (MeteoSwiss, 2024). Further, new high-resolution snow climatologies produced by both MeteoSwiss and the WSL Institute for Snow and Avalanche research SLF have recently become available, providing a novel opportunity to analyze the long-term influence of snow processes, which are crucial for streamflow (drought) generation in Alpine catchments in Switzerland (Staudinger et al., 2014, 2017; Avanzi et al., 2024; Brunner et al., 2023; Koehler et al., 2022; Michel et al., 2023; Marty et al., 2025).

Observation-based evapotranspiration and soil moisture data is sparse in Switzerland. Hence, information on these variables is often extracted from hydrological model simulations Brunner et al. (2021); Melsen and Guse (2019); Samaniego et al. (2013, 2018). The ERA5-Land reanalysis dataset, provided by the European Centre for Medium-Range Weather Forecasts



(ECMWF) (Muñoz-Sabater et al., 2021), offers a compromise between high spatial resolution and long temporal coverage and is better suited for hydro-meteorological analyses and modelling over more complex terrain such as Switzerland than the ERA5 reanalysis datasets (Muñoz-Sabater et al., 2021). A frequently used approach for analyzing drought propagation from meteorological (precipitation) to agricultural (soil moisture) and hydrological (streamflow and/or groundwater) droughts relies on standardized drought indices based on e.g., precipitation and/or evaporation (by using the standardized precipitation index

(SPI) or the standardized precipitation evaporation index (SPEI) (Raposo et al., 2023; Barker et al., 2016; Peña-Gallardo et al., 2019; Zhou et al., 2021). These standardized drought indices are typically aggregated over varying retrospective time scales (months to years) and are useful proxies for various factors that determine catchment-scale water balances, including soil moisture, streamflow, groundwater, and snow processes (Bachmair et al., 2018; Tschurr et al., 2020; European Commission, 2020; Cammalleri et al., 2019; Staudinger et al., 2014). Longer aggregation scales hereby reflect response scales of storage

components with longer memory, while shorter scales reflect streamflow and/or soil moisture in smaller catchments, mainly influenced by pluvial processes (Bachmair et al., 2018; Baez-Villanueva et al., 2024; Haslinger et al., 2014; Myronidis et al., 2018; Staudinger et al., 2014; Tschurr et al., 2020; WMO and GWP, 2016; Yihdego et al., 2019; Cammalleri et al., 2019; Bachmair et al., 2016; European Commission, 2020). Standardized drought indices are now widely used in DEWS (Bachmair et al., 2016; Kchouk et al., 2022; Raposo et al., 2023; Tijdeman et al., 2020) and will also be used in the Swiss DEWS (L.

Benelli, pers. comm.).

Recent studies focused on assessing the benefits of non-standardized (deficit) indices in tracking the drought propagation signal across drought types (see e.g., Brunner and Chartier-Rescan, 2024; Sur et al., 2020; Wu et al., 2020). Non-standardized indices provide physically interpretable and consistent information on deficits which remain inter-comparable across systems as a result of non-transformation (Van Loon, 2015; Raposo et al., 2023; Wu et al., 2020). Examples are the Hydrological Anomaly

Index (HAI), the Water Balance Drought Index (WBDI), the cumulative water deficits (CWD), and the potential cumulative water deficit (PCWD) (Stocker et al., 2023; Sur et al., 2020; Wu et al., 2020). Non-standardized indices allow direct quantification of (precipitation) deficits or surpluses associated with the drought propagation into and recovery from a (hydrological) droughts (Wu et al., 2020) and hence provide valuable information for proactive water management and decision-making (Xu et al., 2023; Parry et al., 2018).

Here, we present a novel dataset with high-resolution observational daily catchment-level time series for key hydro-meteorological variables (including precipitation, snow water equivalent, temperature, soil moisture, (potential) evaporation and streamflow), standardized and non-standardized (drought/deficit) indices (SPI, SPEI, SMRI, CWD, PCWD, CQD) and (streamflow) drought events covering 184 small to large catchments in Switzerland. The HYD-RESPONSES dataset can be combined with existing hydro-meteorological time series datasets and catchment descriptors such as CAMELS-CH (Höge

et al., 2023a).

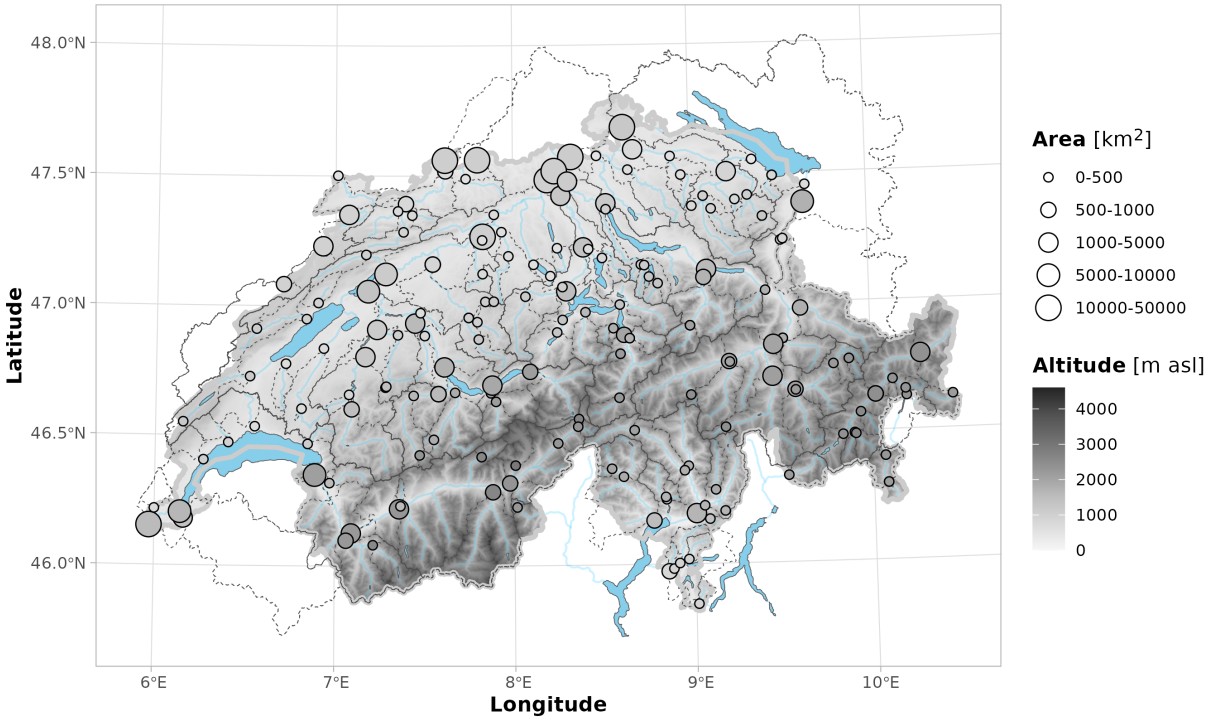

**Figure 1.** Overview of the study area and catchments included in the HYD-RESPONSES dataset. Catchment outlets (circles) are coloured by mean catchment altitude [m a.s.l.] and the point size scales with the catchment area [km$^2$]. Dashed lines show the catchment outlines. Generalized streamflow networks and lakes are shown in light blue.

## 2   Study region and catchments

The 184 catchments (Fig. 1) provided in the HYD-RESPONSES dataset span a wide range of catchment areas (0.56–35'878 km$^2$), glaciation percentages (0–56 %), altitude ranges (467–2937 m a.s.l.) and streamflow regime types (n=18) (see Fig. 3). More than half (n=94 (51 %)) of the catchments are small to mid-sized with an area of between 10 km$^2$ and 500 km$^2$. 9 (4 %) catchments are smaller than 10 km$^2$ and 56 (30.4 %) catchments are larger than 500 km$^2$. The dataset contains eight very large catchments with areas between 10000 km$^2$ and 50000 km$^2$ (max. area = 35'878 km$^2$), associated with the three largest rivers in Switzerland: Aare, Rhine and Rhone. Most catchments (82.5 %) have less than 5 % glaciated area. The catchments are distributed relatively equally between 500 and 2500 m a.s.l. with fewer (77 out of 98) catchments at elevation ranges above 1500 m a.s.l.. Only eight catchments are higher than 2500 m a.s.l. and only one catchment is at very low elevation (catchment Wiese, Basel). Streamflow regime types were classified and adjusted by the FOEN based on data from the Hydrological Atlas of Switzerland Table 5.2 (https://hydrologischeratlas.ch/downloads/01/content/Tafel_52.pdf). Catchments smaller than 500 km$^2$ are characterized by considering mean altitude and catchment glaciation percentage to reflect the contribution of specific streamflow (drought) generating processes (glacial, nival, pluvial). Catchments larger than 500 km$^2$ are generally classified





as *mixed regime (>500 km$^2$)* type and contain catchments characterized by a combination of streamflow (drought) generating

processes. For more information see also Aschwanden and Weingartner (1985) and Fig. 3e.

## 3   Input data products

In this section, the input datasets used to produce and compile the HYD-RESPONSES dataset are presented and reference
literature for further reading and more detailed information is provided. Original data products are provided by the Federal
Office for Climatology and Meteorology (MeteoSwiss), the Federal Office for the Environment (FOEN), the Swiss Federal
Office of Topography (Swisstopo), the Federal Office for Agriculture (FOAG), the WSL Institute for Snow and Avalanche
Research (SLF) and the European Centre for Medium-Range Weather Forecasts (ECMWF).

### 3.1   Catchment-level time series data from streamflow observations

Daily average streamflow measurements at the catchment outlet were provided by the FOEN via the Hydrological Service
(www.hydrodaten.admin.ch) for more than 200 stations. The data availability is station-specific and depends on the installa-
tion and FOEN-internal data quality checking. The HYD-RESPONSES dataset only provides a subset of 184 catchments by
considering only stations for which an analysis of hydrological drought dynamics in response to cumulative water deficits was
deemed to be meaningful in correspondence with the FOEN (Caroline Kan; see Fig. 1). Stations were excluded in case of
i) Q measured at water-level stations (3 stations), ii) Q measured at NADUF-stations (4 stations), iii) secondary stations (11
stations), iv) stations with potential return (= negative) streamflow (2 stations), v) Q measured at derivations (2 stations), vi)
stations with no watershed delineation (i.e., subterranean; 1 station) and vii) uncertainties in time series composition due to
displacement and/or temporarily missing Q of contributing stations (4 stations). A complete list of included stations is provided
in Tables A2, A3, A4 and A6 (Appendix).

### 3.2   Catchment-level time series data derived from spatially gridded products

Meteorological variables (except for evaporation) were assembled from the high-resolution (1×1 km) spatial climate analyses
provided by MeteoSwiss (MeteoSwiss, 2024) (see Table 1). The variables include average 2 m temperature (TabsD), daily
minimum and maximum 2 m temperature (TminD, TmaxD), daily precipitation sums (RhiresD) and daily sunshine duration
(SrelD) (Frei, 2014; Frei and Schär, 1998; MeteoSwiss, 2021a, b, c). The data availability is product-specific and covers the
period 1961–2023 for RhiresD and TabsD and 1971–2023 for the other products (TminD, TmaxD, SrelD). The spatial climate
analyses products used here only cover the Swiss territory, except for RhiresD, which covers catchments located outside
Switzerland, but draining through Swiss territory. Note that RhiresD is not available for catchments covering regions in France
and Italy before 1992 due to limited meteorological station availability and hence limited data reliability (MeteoSwiss, 2021a).
Catchments with a significant area in France or Italy may therefore be handled with care and/or potentially be excluded from
analysis before 1992 (see Section 7).



**Table 1.** (Spatially gridded) products used for the time series extraction

| Dataset | Variables | Period | Spatial resolution | Temporal resolution | Producer |
|---|---|---|---|---|---|
| Spatial Climate Analyses | TabsD, RhiresD TminD, TmaxD, SrelD | 1961–2023 1971–2023 | 1×1 km | daily | MeteoSwiss |
| Snow Climatology for Switzerland (SPASS) | SWECLQMD | 1961–2022 | 1×1 km | daily | MeteoSwiss & SLF |
| Climatological snow data since 1998 (OSHD) | swee, romc | 1998–2023 | 1×1 km | daily | SLF |
| ERA5-Land | tp, t2m, e, pev, smlt, sd, ssr, ro, sro, swvl1, swvl2, swvl3, swvl4 | 1950–2023 | 0.1×0.1°(ca. 9×9 km) | hourly | ECMWF |
| Streamflow time series | Q | Station specific | catchment-level (outflow point data) | daily | FOEN |

Snow water equivalent (SWE) data was compiled from two high-resolution (1×1 km) datasets. The first and main product resulted from the joint research project "A spatial Snow Climatology for Switzerland (SPASS)" by MeteoSwiss and SLF (Michel et al., 2023; Marty et al., 2025). The preliminary version was produced in 2022 and provides modelled and bias-corrected daily SWE data for the period September 1961–September 2022. The spatial extent is restricted to the Swiss territory. The SPASS SWE is based on the daily TabsD and RhiresD products (see above) and makes use of a quantile-mapping
approach. The model is presented in detail in Michel et al. (2023). The second snow product is based on the Swiss Operational Snow-hydrological model system (OSHD) and is provided by the WSL (SLF) (Mott, 2023; Mott et al., 2023). The OSHD data provides information on both SWE and snowmelt runoff for the period 1998– 2022 (Mott, 2023; Mott et al., 2023).

    All other hydro-meteorological variables, including evaporation, potential evaporation, soil moisture and additional vari-
ables already covered by the previously introduced datasets, were extracted from the ERA5-Land reanalysis dataset provided by ECMWF (Muñoz-Sabater et al., 2021). Several variables are therefore covered by multiple source data and are all included in the HYD-RESPONSES dataset to allow comparative analyses between the different data products. Time series covered by multiple data sources include temperature variables (TabsD, TminD and TmaxD from MeteoSwiss, t2m from ERA5-Land), precipitation (RhiresD from MeteoSwiss, precipitation from ERA5-Land), potential and total evaporation (ERA5-Land), sun-
shine duration (SrelD), snow water equivalent (SWE; from SPASS, OSHD, and ERA5-Land), modelled snow melt (from OSHD and ERA5-Land) and streamflow (FOEN). Additional variables extracted from ERA5-Land include four soil water



**Table 2.** Data products used to extract catchment descriptors.

| Dataset | (Extracted) Variables | Producer |
|---|---|---|
| Digital soil suitability maps of Switzerland | soil wetness, soil depth, permeability, water holding capacity, nutrient content and skeletal content | FOAG |
| Hydrogeological map of Switzerland | aquifer type (loose or solid rock), aquifer genesis and aquifer productivity | FOEN |
| Lithological map for Switzerland | dominant rock type classes (loose, sedimentary and crystalline rock) | Swisstopo |
| Springs and swallow holes in karst regions | number of springs (per $km^2$) | FOEN |
| swissALTI3D (DEM) | aspect, slope | Swisstopo |
| swissTLM3D Hydrography | Drainage density | Swisstopo |
| Biogeographic regions of Switzerland | Biogeographic regions | FOEN |
| Catchment metadata | time series availability, breakpoint analysis, area, mean height, outlet coordinates and streamflow regime type | FOEN |

volume levels (swvl), total solar radiation (ssr) and runoff (ro) and surface runoff (sro). For a more detailed description of variables, see the data documentation on Zenodo (von Matt et al., 2025). A glossary of variable abbreviations is provided in Table A1.

The ERA5-Land data is provided at an hourly temporal resolution for the period 1950–2023 and can be accessed via the Copernicus climate data store (CDS) (https://cds.climate.copernicus.eu/datasets/reanalysis-era5-land). We preferably included data from ERA5-Land over data from ERA5 due to the higher spatial resolution of ERA5-Land (0.1×0.1°, ca. 9×9 km). To ensure consistency with the other hydro-meteorological input datasets, the hourly ERA5-Land data was aggregated to daily values (see Section 4.1).

**3.3   Catchment-level time-invariant data (catchment descriptors)**

Datasets used to compile an extensive set of catchment descriptors include station metadata and information on time series availability and homogeneity provided by the FOEN as well as spatial (polygon) data on hydro-terrestrial characteristics (e.g., soil characteristics, hydro-geology) provided by the FOEN, FOAG and Swisstopo (see Tab. 2). Most information is available from www.opendata.swiss, the FOEN Hydro-Service (www.hydrodaten.admin.ch, or can be downloaded and inspected via
www.map.geo.admin.ch (Swisstopo). Direct links to the datasets are provided below in section 10.

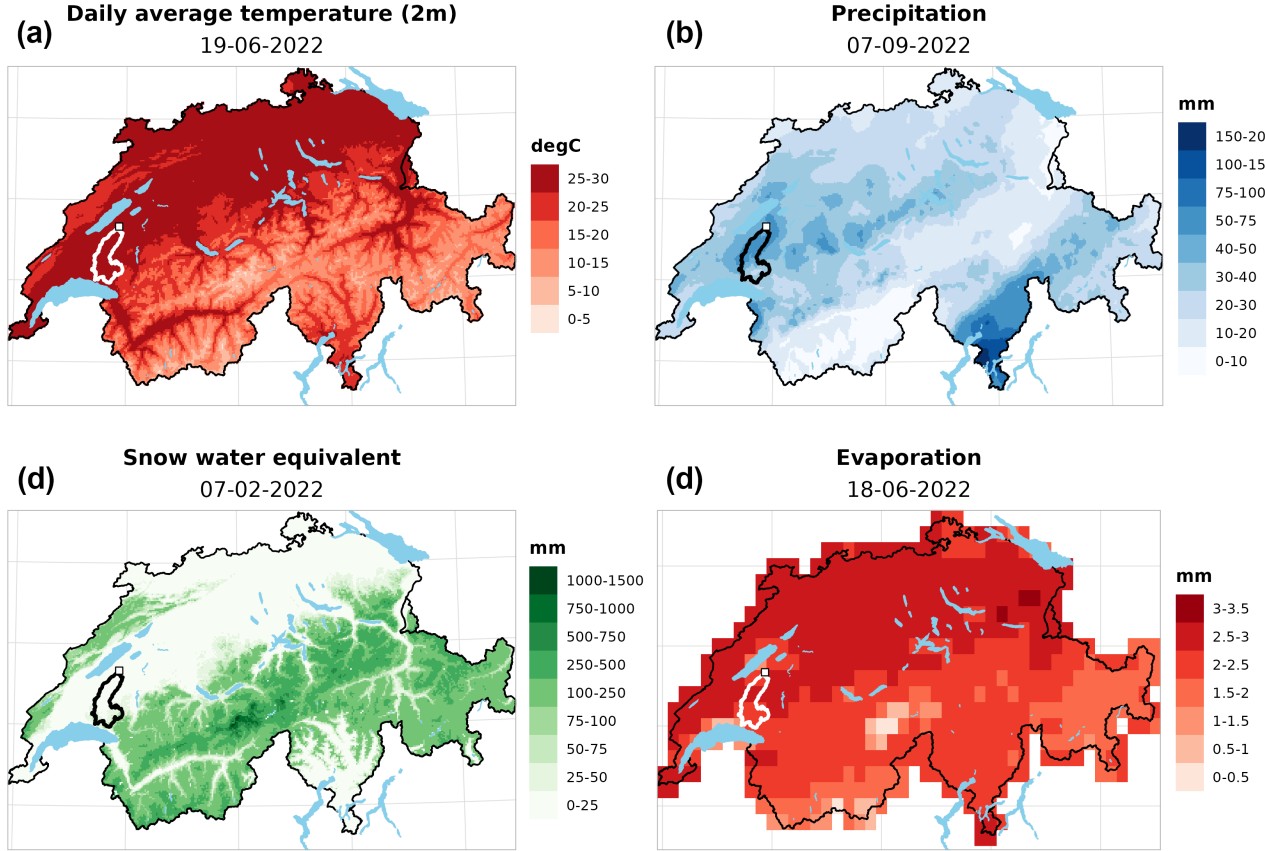

**Figure 2.** Overview of the spatial raster products used to extract daily time series. **(a)** Mean daily temperature (TabsD, MeteoSwiss), **(b)** Daily precipitation sun (RhiresD, MeteoSwiss), **(c)** Daily snow water equivalent of the Swiss snow climatology (SPASS) (SWE, MeteoSwiss & SLF), **(d)** Daily evaporation sum (aggregated from hourly ERA5-Land data, ECMWF). Note that the second snow climatology product (OSHD) is not shown. Contours in white/black show catchment *2034 - Broye, Payerne, Casernde d'aviation* for the day with the highest observed catchment average values for each specific product for the year 2022. White squares show the catchment outlet where daily streamflow is measured. Extracted and derived time series over the year 2022 are shown for the same catchment in Figure 9.

The digital soil suitability maps provide information on a set of different soil characteristics assessed on 25 different geological and geomorphological units which are further discriminated by different landscape elements depending on aspect, slope and bedrock. The maps were first assessed in 1980 and revised in 2000 (BLW, 2022; Swisstopo, 2020). The different soil characteristics include soil wetness, soil depth, permeability, water storage capacity, nutrient content and skeletal content. The hydro-geological map of Switzerland provides information on groundwater resources in Switzerland (Schürch et al., 2007), including information on aquifer type (loose or solid rock), aquifer genesis and aquifer productivity. The map was originally produced and published for the Hydrological Atlas of Switzerland (HADES, https://hydrologischeratlas.ch/). The


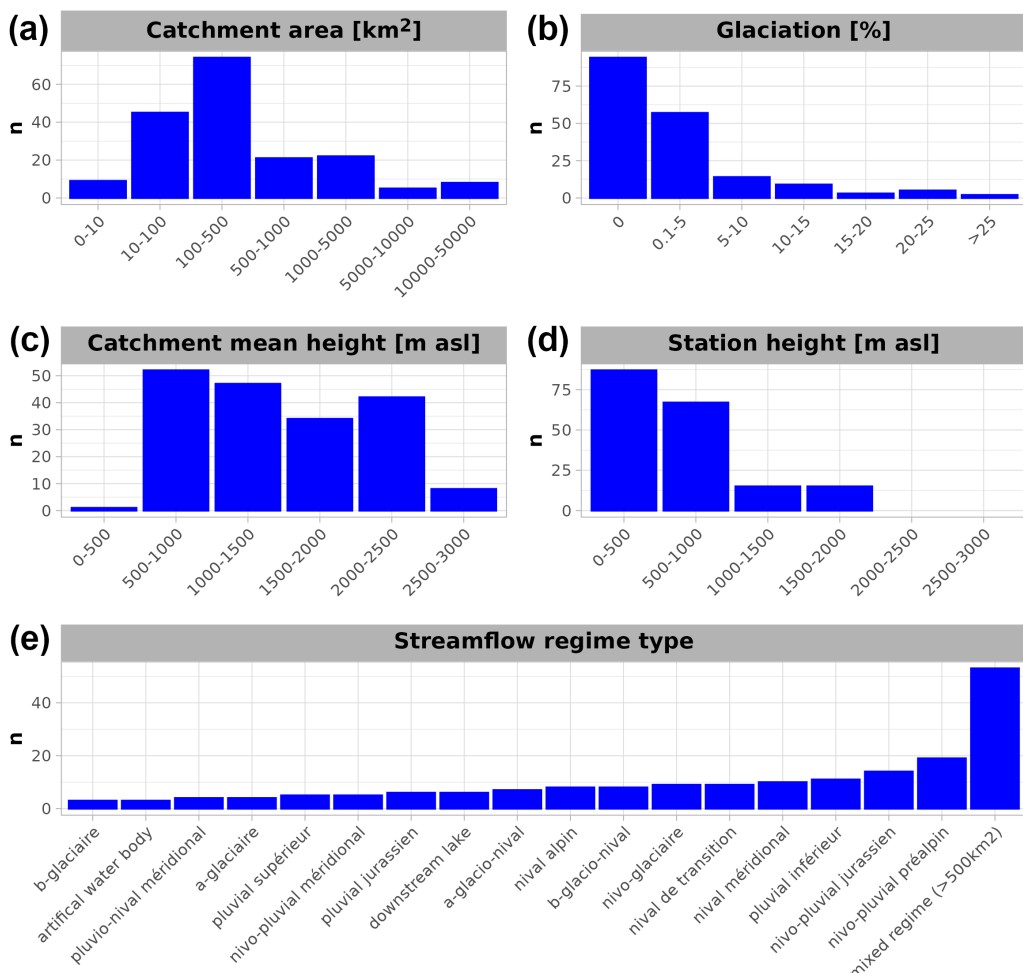

**Figure 3.** General catchment characteristics provided by the FOEN. **a)** Catchment area in km$^2$, **b)** Glaciation percentage (of catchment area), **c)** catchment mean height [m a.s.l.], **d)** height of the streamflow gauge measurement station [m a.s.l.] and **e)** streamflow regime types. The Y-axis shows the frequency of each category.

hydro-geological information was further complemented with the lithological map for Switzerland (produced by Swisstopo), which provides a general overview of dominant rock type classes (loose, sedimentary and crystalline rock). The maps are available via opendata.swiss (hydrogeological map, lithological map) or can also be accessed via the Hydrological Service of the FOEN (https://www.bafu.admin.ch/bafu/de/home/themen/wasser/zustand/karten/geodaten.html). The number of springs and swallow holes in karstic regions provides additional information related to aquifers and the contribution of subsurface water storage. The layer provides main discharge source locations in karstic regions and is available via opendata.swiss





(produced by FOEN). Standard topographical characteristics such as slope and aspect were derived from the high-resolution
digital elevation model (swissALTI3D) publicly available via Swisstopo at a resolution of 2 m (Swisstopo, 2022). The
swissTLM3D Hydrography provides topological information on the different water bodies of Switzerland (including flowing
and stagnant waters) and originates from the swissTLM3D dataset provided by and accessible via Swisstopo.

The biogeographic regions of Switzerland provide six regions differentiated by similarity of flora, fauna, bryophytes and
ornithological information as well as homogeneous surface water catchments (BAFU (Hrsg.), 2022). Biogeographic (eco-
)regions often correspond well to catchment groups with similar streamflow regime types and are therefore frequently used for
catchment regionalization (e.g., Jehn et al., 2020; Guo et al., 2021). The biogeographic regions are available via opendata.swiss.

Finally, general information on the gauging stations and streamflow time series (availability and homogeneity) were provided
as accompanying (meta-)data by the FOEN. Time series homogeneity was assessed by a FOEN-internal breakpoint analysis
for time series homogenization (for more information see BAFU, 2024). General station information includes catchment area,
mean height, glaciation percentage, outlet coordinates and streamflow regime type (among others) (see Figs. 1 and 3). Catch-
ment outlines (polygons provided by the FOEN) and catchment outlets (point shapes) are provided in the coordinate system
CH1903/LV03 (EPSG:21781).

**4    Data processing**

This section describes the methodology used for aggregating spatially gridded data products and catchment descriptors on the
catchment level, the methods used to derive additional indicators, standardized drought indices, and presents the definition and
declaration of (hydrological) drought events.

**4.1    Time series extraction**

Based on the spatially gridded hydro-meteorological input products (see Section 3.2), catchment-level time series were
extracted using the R-packages *terra* (Hijmans, 2023) and *exactextractr* (Baston, 2023). First, the hourly ERA5-Land data
was aggregated to daily resolution following the standards used by the MeteoSwiss spatial climate analyses (e.g., RhiresD
and TabsD). For this, instantaneous and accumulation/flux variables are distinguished. For instantaneous variables, we provide
daily average values. For accumulation and flux, we provide variables daily sums. Flux variables (mainly precipitation and
evapotranspiration) were further aggregated consistently with RhiresD precipitation sums, i.e., from 06 UTC (day) to 06 UTC
(day + 1) (see MeteoSwiss, 2021a). Instantaneous variables and ERA5-Land temperature were averaged from 00 UTC to 00
UTC, which is consistent with the other MeteoSwiss products (e.g., TabsD; MeteoSwiss, 2021b). Daily catchment-average
time series were then extracted by using the catchment outlines (polygons) provided by the FOEN. Units were homogenized
across time series. The units are listed in Table A1.






The length of the time series depends on the dataset that they were derived from (see Table 1 for details). Streamflow time series are provided for three different catchment-specific time periods: 1) the original time series (entire period), 2) the most recent gap-free time-period time series and 3) the most recent homogeneous time series (in case of significant breakpoints; otherwise equal to the gap-free time series) (see Fig. 4). The breakpoint information is provided by FOEN (for more information see BAFU, 2024). Information on the start of the streamflow monitoring by limnographs is also provided. The streamflow data should only be considered reliable after the initialization of a limnograph. In case of no breakpoints the gap-free period is equal to the homogeneous period. The homogeneous period is usually the shortest (e.g., in case of breakpoints or limnograph initialization; see for example catchment 2349 in Fig. 4). In the case of gaps but no breakpoints, both the homogeneous and the gap-free periods are identical (see, i.e., 2239, 2386 and 2368 in Fig. 4). Indicators and (non-)standardized (drought/deficit) indices derived from the hydro-meteorological time series are available for the longest common period of all contributing variables.

### 4.2 Derived indicators

#### 4.2.1 Streamflow

Derived indicators related to streamflow consist of the 7-day average streamflow (moving average) M7Q. The M7Q (or M7) is often used in low-flow studies and is also used for the official low-flow statistics in Switzerland by the FOEN (see e.g., BAFU, 2024; Muelchi et al., 2021a; von Matt et al., 2024).

#### 4.2.2 Snow related variables

In addition to variables providing direct information on (modelled) snowmelt, also daily differentiated SWE ($\Delta$SWE) time series are provided for both SPASS and OSHD. Snowfall ($\Delta$SWE $> 0$) and snowmelt ($\Delta$SWE $< 0$) time series are provided separately. Note that the SPASS SWE is reset at the end of every snow year (every September $1^{st}$) to avoid unrealistically high snow water equivalent accumulation ("snow towers") (Michel et al., 2023). This can result in large snowmelt amounts ($\Delta$SWE $< 0$) around September $1^{st}$. $\Delta$SWE values on September $1^{st}$ were therefore replaced by a linear interpolation between the day before and the day after. Snow-corrected precipitation series (P+ $\Delta$SWE) were calculated by combining time series of total precipitation (RhiresD and ERA5-Land) and $\Delta$SWE time series (SPASS, OSHD) as well as time series with modelled snowmelt information (SPASS, OSHD and ERA5-Land). Negative snow-corrected precipitation amounts (e.g., RhiresD $< \Delta$SWE) were set to zero.

#### 4.2.3 Water balance

(Potential) Water balance indicators (P–E and P–PET) were derived by combining the total and snow-corrected precipitation time series with the ERA5-Land evaporation and potential evaporation time series.





### 4.3 Cumulative water deficits

Cumulative (potential) water deficits (CWD and PCWD) are non-standardized indicators tracking evaporation-driven deficits in the (potential) water balance. CWD and PCWD were derived from the daily water balance indicator time series (see Section 4.2.3) using the *cwd* R-package (Stocker et al., 2023; Stocker, 2021). A deficit starts when the water balance is negative (i.e., $P - E < 0$) and is accumulated as long as the deficit remains uncompensated (deficit $> 0$). Note that no surplus information is tracked. Once the deficit is compensated, the values remain at zero (CWD $= 0$). In some cases, PCWDs (especially for P–PET based only on ERA5-Land variables) are not compensated each year and can persist over multiple years. Both CWDs and PCWDs are hence also provided on a yearly calculation basis (annual reset on December $31^{st}$). Non-standardized indices preserve units (here millimetres) and are physically interpretable in terms of absolute deficit amounts. Cumulative water deficits do not rely on a predetermined calculation time window, which allows the user to track both deficits accumulated over short periods (below one month) and deficits accumulated over very long periods.

### 4.4 Standardized (drought) indices

Standardized (drought) indices depict the anomaly of a deficit over a fixed retrospective period (e.g., 1 month). The hydrometeorological indicator time series is first aggregated over the given period and then transformed to a standard normal distribution by fitting a suitable candidate distribution (Tijdeman et al., 2020; Stagge et al., 2015). Standardized indices therefore provide information on both anomalously dry and wet conditions, which are often defined by thresholds corresponding to standard deviations (STD). As such, values below $-1$ STD indicate drier than normal conditions (moderate droughts), while values above $+1$ STD indicate wetter than normal conditions (moderate wetness) (McKee et al., 1993; Tschurr et al., 2020). The HYD-RESPONSES dataset provides daily time series for three standardized (drought) indices: the Standardized Precipitation Index (SPI, McKee et al., 1993), the Snowmelt and Rain Index (SMRI, Staudinger et al., 2014), and the Standardized Precipitation Evaporation Index (SPEI, Vicente-Serrano et al., 2010). SPI and SMRI represent precipitation-driven deficits, as they are based on total (SPI; P only) or snow-corrected (SMRI; P$+ \Delta$SWE) precipitation time series. The SPEI accounts for deficits driven by evaporation and is derived from the potential water balance (P–PET). Daily time series for all three indices (SPI, SPEI, SMRI) are provided for aggregation periods ranging from 1–24 months (31–730 days).

All indices were calculated using the *SCI*-package (Stagge et al., 2015; Gudmundsson and Stagge, 2016) with custom modifications accounting for the daily time series resolution. All candidate distributions provided within the *SCI*-package (*gamma, genlog, gumbel, lnorm, norm, gev, pe3, weibull*) were tested for suitability. The distributions were fitted for each day of the year (DOY) based on the reference period 1991–2020. The suitability of candidate distributions was assessed based on three indicators: the Shapiro-Wilks normality tests (*p*-values; Shapiro and Wilk, 1965), the number of flags returned by the fitting function (usually indicating convergence issues), and the number of missing and/or implausible values. Implausible values are defined as values above or below $+3$ ($-3$) STD following Stagge et al. (2015). As in Staudinger et al. (2014), one best-fitting distribution is chosen for all catchments and to allow for catchment comparability. The distribution was selected among the distributions satisfying the following conditions: 1) the transformed values are not significantly different from a normal dis-





tribution for the majority of catchments ($p$-values$> 0.05$ for at least 75 % of the catchments), 2) fewer than 5 DOYs flagged and 3) fewer than 50 implausible and/or missing values. The distribution selection procedure is illustrated for the SPEI in Fig. 5. The results of the Shapiro-Wilks tests ($p$-values) and information on missing/implausible values and flags are also provided in the HYD-RESPONSES dataset and can be used to identify catchments with non-satisfying properties within the overall best-fitting distribution (see Fig. 5).

The *Gamma* distribution was chosen for the SPI for all variables (RhiresD, ERA5-Land), which is consistent with other studies and WMO recommendations (WMO and GWP, 2016; Stagge et al., 2015; Tschurr et al., 2020; von Matt et al., 2024). The SMRI was fitted by the *genlog* (*lnorm*) distribution for the snow-corrected precipitation series based on SPASS (ERA5-Land and OSHD). For the SPEI, the *genlog* distribution was found to perform best across time scales (see Fig. 5). Following Stagge et al. (2015), values of all standardized (drought) indices time series were restricted to the interval [-3, 3] STD.

### 4.5 Climatology & Anomalies

Climatologies and anomalies are provided for all time series including the standard time series of extracted variables (see sections 3.1 and 3.2, derived indicators (Section 4.2), standardized (drought) indices (Section 4.4) and cumulative water deficits (Section 4.3 and 4.6). Both climatologies and anomalies are based on the reference period 1991–2020. The climatology is provided for two variants: i) using moving windows and ii) for fixed periods. The variants are available at the following time scales: daily (only i), monthly (both), seasonal (both), and annual (only ii). The moving window climatology was calculated by using a moving window of 31 days ($day - 15$, $day = 0$, $day + 15$) for the monthly, a 3-month window (91 days) for the seasonal and a 6-month (183 days) window for the extended season time scale. The moving window climatology is calculated for DOYs 1–366 with NA-values set for February $29^{th}$ in the case of non-leap years. The regular climatology is available for monthly, seasonal (DJF, MAM, JJA, SON), extended season (Mai–October, November–March) and annual time scales. Using the moving window climatology, standardized anomalies have been derived by calculating z-scores ($(value - \mu)/\sigma$). The following climatological statistics are provided: minimum, maximum, mean, median, standard deviation, $5^{th}$, $25^{th}$, $75^{th}$ and $95^{th}$ percentiles. For the 7-day average streamflow series (M7Q) we also provide the $2^{nd}$, $10^{th}$ and $15^{th}$ percentiles.

### 4.6 Cumulative streamflow deficits

Time series of cumulative streamflow deficits (CQD) were calculated based on negative streamflow anomalies (drought phases) by using the same procedure as for cumulative water deficits (see Section 4.3). CQD time series are provided for both fixed and variable threshold definitions. For the fixed threshold definition, daily M7Q anomalies were derived for the yearly Q347-threshold events ($\approx$ the yearly $5^{th}$ percentile, see Section 5.2). For the variable threshold definition, daily M7Q anomalies were calculated for the following monthly (31 days) and seasonal (91 days) percentiles: $2^{nd}$, $5^{th}$, $10^{th}$, $15^{th}$, $25^{th}$, $50^{th}$ (median) and mean. Cumulative deficits are physically interpretable and in the case of cumulative water deficits [mm] and streamflow deficits [m$^3$/s] also physically comparable in terms of total runoff depth [mm].



## 4.7 Identification of drought events

We define drought events as coherent phases of non-zero deficits for cumulative deficits (CWD, PCWD and CQD) and as negative M7Q-based streamflow anomalies for streamflow droughts. Streamflow drought phases were extracted for the same percentiles and time scales as used for CQDs (see section 4.6), namely for monthly (31 days) and seasonal (91 days) percentiles: $2^{nd}$, $5^{th}$, $10^{th}$, $15^{th}$, $25^{th}$, $50^{th}$ (median) and the mean. For each event definition, the event time series consists of consecutively numbered event phases and information on the event duration since the start. A minor pooling for hydrological drought events is

introduced by using 7-day average streamflow (M7Q) (Tallaksen and Van Lanen, 2004; Hisdal and Tallaksen, 2000; Tallaksen et al., 1997; Sarailidis et al., 2019).

## 5 Catchment descriptors

Catchment descriptors were extracted from spatial datasets containing information on hydro-terrestrial characteristics (e.g., soil suitability maps), catchment (station) metadata (see Section 3.3) and the extracted hydro-meteorological time series (e.g.,

climatology; see Section 4.5). All catchment descriptors provide only static (time-invariant) catchment information. Catchment descriptors are provided as single-value catchment-level information.

### 5.1 Extraction of catchment descriptors

Spatially non-overlapping polygon datasets (e.g., soil suitability maps) typically provide categorized values for variable-specific classes (e.g., soil depth classes are *shallow*, *medium*, *deep*, *very deep*). To extract catchment-level information, polygon-

based information was first rasterized to a spatial grid identical to the MeteoSwiss spatial climate analyses grid products (in both extent and resolution). The rasterization was done by using the *rasterize* function of the *terra* R-package (Hijmans, 2023). Each grid cell only contains the value of the category with the largest overlap. Slope and aspect values were derived from the swissALTI3D digital elevation model (DEM, see section 10 for a download link) by using the standard *terra* R-package function *terrain* (Hijmans, 2023). A custom categorization was then applied to the resulting grid values (e.g., for slope 0–30,

30–60, etc.). The percentage overlap with the catchment area was then assessed for all variable-specific classes by using the *exact_extract* function (as for time series) and adjusting the aggregation function to fractions ("frac"; see Baston, 2023). Catchment area overlap fractions are provided for all categories. Descriptors with multiple classes can also be reduced to a single dominant category represented by the largest percentage overlap ("proportion"). An example is shown for the biogeographic regions in Fig. 7. However, the class with the largest overlap does not necessarily correspond to the most representative, as

multiple categories can share similar proportions of the catchment area.

### 5.2 Derivation of other catchment descriptors

Additional catchment descriptors were derived from the remaining descriptive input products (catchment metadata, karstic sources, catchment outlines and hydrography) and the calculated hydro-climatology (see Section 4.5).





Two descriptive variables related to catchment shape and drainage were derived in R by using the catchment outlines, namely
the *basin shape index* (BSI) and *drainage density*. The HYD-RESPONSES dataset provides two BSI variants. The first variant
is derived based on a ratio between area and length ($A/L^2$) and the second variant is based on a ratio between the catchment area
and the area of the circle with the smallest radius encircling the entire catchment ($A_{catch}/A_{circle}$). For more information see
Das et al. (2022). The drainage density denotes the ratio between the catchment area and the total length of streamflow channels
(both natural and stormwater drainage infrastructure; Dingman, 1978; USGS, 2023). The drainage density was calculated by
using the swissTLM3D hydrography dataset (see section 10 for a download link). Both indices (BSI and drainage density) are
frequently used in flood-related studies but may also provide valuable information during low-flow periods as high-intensity
precipitation events are a relevant factor for (streamflow) drought recovery (Eekhout et al., 2018; Floriancic et al., 2022; Lee
and Ajami, 2023; Matanó et al., 2024; Qiu et al., 2021; Tarasova et al., 2024; Vicente-Serrano et al., 2022; Wu et al., 2022;
Xu et al., 2023). Further, also the overlap percentage with the Swiss territory (swissBOUNDARIES3D, see section 10 for a
download link) is provided for each catchment and can be used to exclude catchments with significant portions outside of
Switzerland which goes along with a limited coverage in both hydro-meteorological and catchment descriptor input datasets
(see Sections 3.2 and 3.3). Information on karstic sources is provided as the number of sources per catchment and $km^2$.

Several indices related to streamflow characteristics (low flow, responsiveness, baseflow and flow stability) are provided in
the HYD-RESPONSES dataset. The Q347 (Aschwanden, 1992; Aschwanden and Kan, 1999) is a low flow index used as the
basis for water abstraction restrictions in Switzerland and corresponds to the $5^{th}$ streamflow percentile (low flows) derived
from the flow duration curve (FDC). The Q347 was derived by using the *hydroTSM* R-package (Zambrano-Bigiarini, 2020).
The baseflow index (BFI; Nathan and McMahon, 1990) is a widely used index linked to multiple catchment characteristics
such as aquifer type, productivity and soil characteristics. The BFI provides information on the (base-)flow sustained during dry
periods (e.g., by subsurface storages; Tallaksen and Van Lanen, 2004; Bloomfield et al., 2021; Van Loon and Laaha, 2015). The
BFI was derived using the *baseflow* function of the *lfstat* R-package (Laaha and Koffler, 2022) and is shown in Fig. 7. Stoelzle
et al. (2020) introduced the delayed-flow index (DFI) which breaks down the BFI into individual hydrograph components. The
components include fast, intermediate, slow and base responses and potentially reflect various storage processes contributing to
the overall streamflow response (e.g., snowmelt and groundwater). The DFI was derived by using the *delayedflow* R-package
(https://modche.github.io/delayedflow/; see also Stoelzle et al., 2020). The last two indices related to streamflow behaviour
are the "flashiness" or R-B-index (Baker et al., 2004) which represents the ratio of the sum of day-to-day streamflow changes
divided by the total streamflow and the flow-stability index which relates the mean annual minimum flows to the mean annual
flow (MAM/MQ).

The remaining catchment descriptors were derived from the extracted hydro-meteorological time series and/or their respective
climatology. Information on average precipitation, temperature, evaporation, snow water equivalent, streamflow, the fraction of
precipitation falling as snow and the runoff fraction (Q/P) are provided, partly on both monthly and yearly time scales. Finally,
monthly Pardé coefficients (PCs) are provided which indicate the contribution of monthly mean streamflow to the annual mean
streamflow.





## 6 Three example use cases

The different data types can be combined to comprehensively analyse hydrological streamflow droughts in response to various
hydro-meteorological indicators. This section presents three use cases: catchment regionalization, in-depth event analysis, and composite analysis. A comprehensive R-tutorial on how to read and combine the different data products is provided with the dataset but can also be accessed via Github (https://github.com/codicolus/HYD-RESPONSES).

### 6.1 Catchment grouping

For some applications, catchments need to be grouped by similarity, as measured by a set of hydro-meteorological, terrestrial
and/or anthropogenic catchment descriptors (e.g., Tarasova et al., 2024). As an example application, we show the distribution of catchment coverage fractions across biogeographic regions for the soil characteristics *soil depth, skeletal content, water logging, permeability, and water storage capacity* (Figure 8).

The percentage coverage distributions reveal notable differences in soil characteristics and their subcategories. Catchments in the Swiss Plateau region are characterized by larger coverages of deep to very deep soils with a mostly poor to medium
skeletal content, normal soil permeability and good water storage capacities (see Fig. 8). Alpine catchments, on the other hand, are characterized by shallower soils (especially the Southern Alps) and a higher skeletal content. Soils in the Alps further have almost no water logging and a low water storage capacity. Soils with a (weakly) inhibited permeability or with a very good water storage capacity are infrequent across all biogeographic regions. An other example for catchment grouping is the streamflow regime type classification for Switzerland (see e.g., Aschwanden and Weingartner, 1985; Weingartner and Schwanbeck, 2020).
Figure A1 in Appendix A shows the incidence of streamflow regime types across biogeographic regions.

### 6.2 Detailed Event analysis

The combination of hydro-meteorological indicators, standardized (drought) indices (SPI, SPEI, SMRI), cumulative (potential) water (balance) and streamflow deficits (CWD, PCWD, CQD) and accompanying climatological anomalies allow for a detailed analysis of specific (streamflow) drought events. Drought-generating processes vary across catchments depending
on hydro-climatological and terrestrial catchment characteristics, the season as well as on anthropogenic disturbances (e.g., Brunner et al., 2022; Van Loon and Van Lanen, 2012; Van Loon, 2015; Apurv et al., 2017). Except for glacier melt and groundwater, the HYD-RESPONSES dataset provides time series for all relevant hydro-meteorological indicators required to analyse (streamflow) drought generation, drought propagation as well as drought type classification.

Figure 9 illustrates time series for the year 2022 of a subset of relevant hydro-meteorological variables for catchment *2034 -*
*Broye, Payerne, Caserne d'aviation*. This catchment is located in the western Swiss Plateau region (highlighted in Fig. 2). The year 2022 was an exceptional year with unprecedented combined heat and drought conditions over Europe (Tripathy and Mishra, 2023). The Broye catchment experienced low-flow conditions beyond a 100-year return period (BAFU (Hrsg.), 2023). In the Broye catchment, the lowest 7-day average streamflow values were observed between July and August (see Fig. 9i) Several streamflow drought events were identified for both yearly fixed (purple shading) and variable (green shading)



threshold definitions. The longest events occur during the annual low-flow season for both definitions.

The year 2022 was also one of the warmest years on record with three heatwaves occurring in mid-June, mid-July and in the beginning of August (Imfeld et al., 2022). During the longest streamflow drought event in July 2022 (M7Q row in Figure 9i), evaporation anomalies begin to decline and become negative towards the end of the event (ET row in Fig. 9c). Concurrent strong negative soil moisture anomalies at shallow and deeper levels (see Fig. 9d) suggest that the successively decreasing

evaporation anomalies may be related to increasingly depleted soil moisture storages resulting in limited water availability for evaporation. Interactions between (subsurface) storage processes are however complex and also include groundwater–soil moisture interactions (e.g., Orth and Destouni, 2018).

The HYD-RESPONSES dataset further provides information on cumulative (atmospheric) water deficits represented by standardized and non-standardized (drought) indices. For the Broye catchment, the 2022 streamflow drought events identified

with the variable threshold (green shading) correlate well with shorter aggregation scales (1- to 3-monthly) SPI and SMRI indices in spring and summer. The correspondence between short-term precipitation deficits and streamflow droughts is, however, not consistent throughout the year. During the variable threshold streamflow droughts in mid-March to April, both SMRI-1 and SMRI-3 reach more negative values than their SPI equivalents, which suggests a contribution of lacking snowmelt to the streamflow drought generation (see Fig. 9g,h).

Cumulative deficits in actual (CWD, Fig. 9k) and potential (PCWD, Fig.9l) water balance as well as streamflow (CQD, Fig. 9j) provide complementary information to the SPI, SMRI and SPEI in the form of non-standardized and hence physically interpretable deficit amounts. Cumulative streamflow deficits (CQD) show only two phases without deficit compensation for both drought definitions (Fig.9j). A shorter CQD phase coincides with the drought events in spring (variable threshold) and the shorter drought event in June (fixed threshold), while a longer phase coincides with the remaining shorter and longer streamflow

drought phases in July and August before CQD is compensated by September 2022. For both CQD phases, the CQD is larger for streamflow droughts based on a variable threshold definition. Above average precipitation (+130 %; BAFU (Hrsg.), 2023) was reported in September 2022 and corresponds well with the compensation of CQD and is also reflected in the positive monthly (31d) precipitation anomalies (P anomaly, Fig. 9b). Similar to the longest streamflow drought phases, also the largest deficits in (actual) water balance (CWD) occurred between May–August 2022. Larger CWDs during the warm season are consistent with

the seasonal climatology of both temperature and evaporation with the highest values during summer (not shown). Major CWD phases match streamflow drought phases remarkably well, especially for the variable threshold definition with one exception in April. The two longer streamflow drought phases in May–June further show the benefits of considering anomalies in the CWDs. While absolute CWDs were not compensated in between the streamflow droughts, the CWD anomalies indicate that the deficits returned to seasonal norm values (see Fig. 9m). Cumulative deficits in potential water balance (PCWDs, Fig. 9l) are

more similar to cumulative streamflow deficits (CQD) for the variable-threshold definition. This reflects the different nature of CWDs and PCWDs. The actual water balance is more strongly tied to the actual water availability and hence the individual streamflow phases. The potential water balance, on the other hand, represents the deficit that would have been accumulated under unlimited water availability. Similar to PCWD, CQDs reflect the integrated streamflow deficit over time while an actual deficit in terms of low streamflow levels does not necessarily have to exist (anymore).





## 6.3 Composite analysis (catchment response patterns)

Composite analysis is a frequently used approach to understand the driving processes of a phenomenon such as droughts (see e.g., Bevacqua et al., 2021; Floriancic et al., 2020; Mahto and Mishra, 2024). By considering the median values of drought indicators across all streamflow drought events in a catchment, typical response patterns may become more evident and may allow for more generalized inferences on typical streamflow drought response patterns e.g., to precipitation deficits accumulated over various aggregation time-scales. Here, we present a composite analysis of median SPI values associated with streamflow droughts defined by the monthly $15^{th}$-percentiles of the streamflow.

Note that streamflow drought characteristics and drought propagation processes may differ among catchments depending on hydro-meteorological climatologies, geological and terrestrial characteristics (e.g., aquifer, rock type, (soil) water storage capacity), seasonality of and differences in contributing streamflow (drought) generating processes and human disturbances (e.g., Van Loon and Laaha, 2015; Floriancic et al., 2022; Jehn et al., 2020; Apurv and Cai, 2020; Savelli et al., 2022; Haile et al., 2020; Brunner et al., 2022, 2021, 2023; Tijdeman et al., 2022; de Jager et al., 2022). We therefore separate the streamflow droughts and catchments by seasons winter (DJF, December–February), spring (MAM, March–May), summer (JJA, June–August) and autumn (SON, September–November) and by streamflow regime types. Six streamflow regime types are selected to capture a variety in dominant streamflow (drought) generating processes. These include glacial (*a-glaciaire, nivo-glaciaire*), nival (*nival alpin, nival méridional*) and pluvial (*pluvial jurassien, pluvial supérieur*) processes. The importance of precipitation deficits across scales is assessed using SPIs (SPI-1 to SPI-24). Streamflow drought events are only considered for the longest common homogeneous period across catchments (1991–2022). The selection is further restricted to catchments with at least 10 streamflow drought events in each season (over the entire time series length) with a minimum duration of at least 10 days to enhance robustness and exclude minor droughts.

Median SPI values are mostly negative across all aggregation time-scales indicating that precipitation conditions co-occuring with streamflow droughts tend to be drier than normal. Several streamflow regimetype-specific response patterns are evident and change across seasons along with contributing streamflow (drought) generating processes.

Glacier melt is the dominant factor for the *a-glaciaire* regime type. Streamflow levels are typically lowest in winter (January–March) as a result of precipitation falling as snow (intermediate storage) and highest in summer due to large contributions of glacier melt (Aschwanden and Weingartner, 1985; Weingartner and Schwanbeck, 2020; Muelchi et al., 2021b). Streamflow droughts of strongly glaciated catchments are not associated with moderate drought conditions at any SPI scale. In glacial and nival catchments a shift towards short-term precipitation deficits (SPI-1 to SPI-6) being associated with droughts is present across seasons and drought-generating processes. The transition towards shorter deficit scales emerges in summer for nival regime types and in autumn for glacial regime types. In pluvial and transitional regime types short-term precipitation deficits (mostly 1- to 3 months) are relevant throughout the year. Seasonal shifts are also observed for pluvial and transitional regime types with mid- and long-term precipitation deficits becoming more relevant in summer and autumn.

In addition to 3-monthly precipitation deficits, also mid- and long-term deficits become relevant in summer and autumn for (nivo-pluvial) catchments in the Jura region and catchments of the regime type *pluvial inférieur*. Compound moderate droughts





are mainly observed for sub-yearly (1- to 9-monthly) scales with most extreme conditions on a 6-monthly scale in the Jura
region (especially for nivo-pluvial catchments) and on a (6- to) 9-monthly scale for catchments of the regime type *pluvial
inférieur*. In southern Switzerland, precipitation deficits tend to be relevant on longer scales compared to similar regime types
north of the Alps. In contrast to nival catchments north of the Alps (*nival alpin*), droughts in the nival catchments south of
the Alps (*nival méridional*) are associated with substantial precipitation deficits at longer aggregation times (9–24 months).
The deficits occur in winter and in summer, but conditions are more extreme in summer (SPI ≈ -1.5) on scales longer than
15 months. Further, also 3-monthly precipitation deficits appear to be relevant for streamflow (drought) generation in summer
(moderate drought conditions). In spring and autumn, mid- to short-term accumulation scales are more relevant. Interpretations
of the differences between the south and north sides of the Alps should however be considered with caution due to the small
catchment sample sizes and the spatial proximity of the two *nival méridional* catchments. The observed response patterns may
therefore not be representative of nival catchments south of the Alps in general.

## 7  Discussion

The HYD-RESPONSES dataset can, for example, be used to study drought dynamics and drought propagation, for streamflow
forecasting using Long Short-Term Models Kratzert et al. (LSTMs; 2018); Lees et al. (LSTMs; 2022); Kratzert et al. (LSTMs;
2023), Random Forests Floriancic et al. (RFs; 2022), or to infer drought drivers using clustering and principal component
analysis (e.g., Jehn et al., 2020). The information on breakpoints allows to study pre- and post-influence catchment behaviour
e.g., by using the paired catchment or upstream-downstream approach (see e.g., Rangecroft et al., 2019; Van Loon et al.,
2019). The availability of variables originating from multiple data sources (direct observations, reanalysis data, model data)
allows for comparative analyses. The following variables are available from multiple sources: temperature (MeteoSwiss,
ERA5-Land), precipitation (MeteoSwiss, ERA5-Land), and snow (MeteoSwiss, WSL, ERA5-Land).

There are several known limitations related to the datasets used to compile the HYD-RESPONSES data. ERA5-Land is a
state-of-the-art reanalysis product provided at a higher spatial resolution than the standard ERA5 reanalysis (Hersbach et al.,
2020; Muñoz-Sabater et al., 2021). The higher spatial resolution results in a better depiction of soil moisture, lakes, river
discharge estimations, and the orographic enhancement of precipitation (Muñoz-Sabater et al., 2021). However, the grid
resolution of 9 km still has limitations over complex high-altitude terrain. The extracted time series related to snow depth
(SWE) should be used with caution, as snow depth in ERA5-Land is of mixed quality depending on geographical location
and altitude (Dalla Torre et al., 2024). Scherrer et al. (2023) showed, that ERA5-Land overestimates SWE at high elevations
with larger biases in the southern compared to the northern Alps. They state that higher-resolution datasets such as SPASS
(Marty et al., 2025) and OSHD (Mott, 2023; Mott et al., 2023) should be preferred over ERA5-Land. Further also note that all
snow-related datasets have problems in representing small SWE amounts at low altitudes (Scherrer et al., 2023; Michel et al.,
2023; Marty et al., 2025).
Another limitation of the ERA5-Land dataset is the parameterization of subgrid-scale processes and the representation of
subsurface storages that affect evapotranspiration (e.g., fixed maximum storage volume assumption; see Muñoz-Sabater et al.,



2021). However, gridded observation-based evaporation datasets are yet to be developed for Switzerland.

Caution is required when using the snow-corrected precipitation (water input) time series. The time series corrected by the $\Delta$SWE series consider both snowfall ($\Delta$SWE $> 0$) and snowmelt ($\Delta$SWE $< 0$), while the correction based on snowmelt

variables only accounts for snowmelt (*smlt* in ERA5-Land and *romc* in OSHD; see Table 1 and Table A1). Snowmelt-corrected precipitation time series only account for snowmelt, they may, therefore, be of limited use during the main snow accumulation season but can still provide valuable information during the snowmelt season.

The time series for standardized (drought) indices are only provided for the transformation based on the best-fitting distribution across all catchments to allow for catchment comparability (see e.g., Staudinger et al., 2014). The best-fitting distribution may

however vary across catchments and climates (see e.g., Stagge et al., 2015). The HYD-RESPONSES dataset provides information on fits, missing values and flags which can be used to exclude catchments with unsatisfying fitting and transformation properties from analyses.

The HYD-RESPONSES time series are provided for product-specific periods and the spatial coverage is restricted to Swiss territory for most of the higher resolution MeteoSwiss and SLF products (TabsD, TminD, TmaxD, SPASS, SrelD, OSHD)

as well as many catchment descriptor input datasets. Full coverage over the entire hydrological Switzerland is only available for ERA5-Land (all variables) and the MeteoSwiss RhiresD product (after 1992; see MeteoSwiss, 2021a). Catchments with significant areas outside of the Swiss National borders may therefore be considered with caution or excluded from the analysis.

## 8 Complementary datasets

Complementary datasets provide a wide range of additional catchment descriptors and hydro-meteorological time series. An overview of datasets and variables is provided in Table 3. The FOEN provides additional geodata related to both surface and groundwater via the Hydrological Service (https://www.bafu.admin.ch/bafu/de/home/themen/wasser/zustand/karten/geodaten. html). The datasets include additional catchment descriptors with information on population density, catchment areas covered by forest and agriculture (among others) as well as information on water quality aspects and sewage. The FOEN further op-

erates both a groundwater monitoring network (NAQUA) providing continuous groundwater measurements for selected point locations (BAFU, 2019) and a water quality measurement network (NAWA) providing information on concentration and loads of important dissolved compounds (e.g., pH, electric conductivity, nutrient contents; BAFU, 2023).

The "Catchment Attributes and Meteorology for Large-sample catchment Studies" (CAMELS) datasets aim at providing a consistent set of hydro-meteorological time series and catchment descriptors over a large sample of hydrological catchments on

country level (Clerc-Schwarzenbach et al., 2024). The catchments in the Swiss version of the CAMELS data (CAMELS-CH; Höge et al., 2023a) are largely congruent with our dataset. The only exception is station 2646, which is only contained in the HYD-RESPONSES dataset. Note that the HYD-RESPONSES dataset provides only a sample subset of 184 catchments. The CAMELS-CH dataset provides valuable complementary catchment-level information on glacier changes (based on GLAMOS, for details see Höge et al., 2023a), land use, hydro-geological and hydro-terrestrial information (e.g., the contributions of vari-



**Table 3.** Datasets compatible and complementary to the HYD-RESPONSES dataset.

| Dataset | Short description | Provider |
|---|---|---|
| Accompanying catchment information | Includes catchment proportions of forests, agricultural (crop) land, population, built-up area and more | FOEN |
| Groundwater measurement network (NAQUA) | Groundwater measurements | FOEN |
| Water quality measurement network (NAWA) | Information on water quality parameters | FOEN |
| CAMELS-CH | Swiss version of the Catchment Attributes and Meteorology for Large sample catchment Studies (CAMELS) dataset | Zenodo |
| MeteoSwiss CombiPrecip (CPC) | High-resolution precipitation fields at ground based on a combination of radar and measurement data | MeteoSwiss |
| HydCHeck | Detailed evaluation of influences and disturbances of the streamflow at NAWA measurement stations | FOEN |

ous grain size categories and bulk-density) as well as anthropogenic disturbances (e.g., hydropower and reservoir capacities). CAMELS-CH further provides modelled time series based on the hydrological model PREVAH (see e.g., Höge et al., 2023a; Viviroli et al., 2009). The CAMELS-CH dataset is freely available from Zenodo (https://zenodo.org/records/10354485; Höge et al., 2023b).

The CombiPrecip dataset (MeteoSwiss) provides high-resolution (10 minutes, 1×1 km) precipitation fields derived from a
combination of radar and station measurement data (Sideris et al., 2014). The CombiPrecip dataset could be a valuable addition for studying drought recovery where extreme precipitation is often considered an important factor (Wu et al., 2022).

The HydCHeck project (Streeb et al., 2024) evaluated the influence of (anthropogenic) disturbance factors on streamflow at stations of the National Surface Water Quality (NAWA) Programme (BAFU, 2023). The evaluated NAWA stations are largely (87.5% of the stations) congruent with the HYD-RESPONSES dataset. The HydCHeck dataset provides catchment-level infor-
mation on the magnitudes for all evaluated disturbance categories including water storage and regulation, hydropower, sewage water, constructions, agriculture as well as drinking and groundwater. The overall impact on several hydrological properties including low-, mid- and high-flow regimes as well as short-term effects and hydraulics is provided as categorical information (from "not disturbed" to "strongly disturbed"). For more information see Streeb et al. (2024).

As part of the planned Swiss National drought early warning system (DEWS), both a high-resolution remote-sensing based
evaporation product (V. Humphrey, pers. comm.) and an automatic soil moisture measurement network are under development at MeteoSwiss, ETH Zurich and WSL and may become a valuable addition in a future.





# 9 Conclusions

The HYD-RESPONSES dataset contains data for 184 Swiss catchments that cover a variety of streamflow regimes, mean altitudes, catchment areas, and anthropogenic influences/disturbances. The catchments cover all biogeographic regions of Switzerland. The HYD-RESPONSES dataset provides daily streamflow data and daily hydro-meteorological time series extracted from gridded data products of MeteoSwiss (TabsD, RhiresD, TmaxD, TminD, SrelD), Meteoswiss and SLF (SPASS), SLF (OSHD) and ECMWF (ERA5-Land). The variables include temperature, precipitation, evaporation, sunshine duration, solar radiation, snowmelt, snow water equivalent, soil moisture, surface runoff, runoff, and streamflow. HYD-RESPONSES further provides derived variables related to streamflow (e.g., M7Q), water balance (e.g., P–E) and snowfall. Additionally, three standardized drought indices (SPI, SPEI, SMRI) for accumulation periods from 1 to 24 months and information on the (non-standardized) cumulative water deficit (CWD), the potential cumulative water deficit (PCWD) and cumulative streamflow deficit (CQD) are provided.

The data set alsop provides information on (streamflow) drought events (occurrence and duration). For each catchment, the drought events have been identified based on fixed and on seasonally varying percentile thresholds.

The combination of data sources, the information on hydro-meteorological variables (mainly temperature, precipitation and snow), the derived indices (water balance, cumulative water deficits, standardized drought indices, climatology and anomalies) allow for a multi-purpose use and various analytical approaches such as time series analysis (e.g., Kratzert et al., 2018; Lees et al., 2022), drought propagation and catchment sensitivity analysis (e.g., based on principal component analysis and clustering; Jehn et al., 2020) and changes in rainfall-runoff relationships during hydrological droughts (e.g., Wu et al., 2021). The HYD-RESPONSES data set can easily be combined with complementary datasets such as CAMELS-CH (Höge et al., 2023a) and HydCHeck (Streeb et al., 2024). The catchment time series vary in length (subject to station initialization), the hydrological time series are provided for the entire measurement period along with information on data homogeneity (see BAFU (2024) for more details).

Limitations exist for catchments extending beyond the Swiss borders. The catchment descriptors were extracted from datasets only covering Swiss national territory. The MeteoSwiss-based datasets cover only Switzerland except for RhiresD, which covers the entire hydrological Switzerland from 1992 onward. In summary, the data set provides a state-of-the-art data basis to study droughts in Switzerland.

*Code and data availability.* The HYD-RESPONSES dataset is freely available (CC BY 4.0) from Zenodo (https://doi.org/10.5281/zenodo.14713274; von Matt et al., 2025). Regular updates are not planned. An R tutorial on how to use and combine the different data products is provided with the dataset but can also be accessed on GitHub (https://github.com/codicolus/HYD-RESPONSES).

As of now, MeteoSwiss gridded spatial analyses products (MeteoSwiss, 2021a, b, c) are not available for free but will be available for free in the course of 2025 (MeteoSwiss, 2025). The preliminary snow climatology for Switzerland (SPASS; see Michel et al., 2023; Marty et al., 2025) was provided directly by MeteoSwiss and is not yet available for public use. The SLF snow climatology (OSHD; Mott, 2023; Mott et al., 2023) was published under the WSL Data Policy and can be downloaded via



Envidat (https://www.envidat.ch/#/metadata/climatological-snow-data-1998-2022-oshd). The hourly ERA5-Land dataset (Muñoz-Sabater et al., 2021) is accessible via the Copernicus Climate Data Store (CDS) (see https://cds.climate.copernicus.eu/datasets/reanalysis-era5-land?tab=download). Daily streamflow time series can be requested via the Hydrological Service of the FOEN via https://www.bafu.admin.ch/bafu/de/home/themen/wasser/zustand/daten/messwerte-zum-thema-wasser-beziehen.html. The soil suitability maps (FOAG), the hydrogeological map (FOEN) and the lithological map (Swisstopo) are available from https://opendata.swiss or

directly via Swisstopo (https://www.swisstopo.admin.ch/de/geokarten-500-vektor). Directly available from Swisstopo are also the datasets swissALTI3D (https://www.swisstopo.admin.ch/de/hoehenmodell-swissalti3d#swissALTI3D---Download), swissTLM3D Hydrography (https://www.swisstopo.admin.ch/de/landschaftsmodell-swisstlm3d#swissTLM3D---Download) and swissBOUNDARIES3D (https://www.swisstopo.admin.ch/de/landschaftsmodell-swissboundaries3d). Further available via https://opendata.swiss are the Biogeographic regions (https://opendata.swiss/de/dataset/biogeographische-regionen-der-schweiz-ch; see also BAFU (Hrsg.), 2022) and information on

karstic springs and swallow holes (also produced by the FOEN; https://opendata.swiss/de/dataset/quellen-und-schwinden-in-karstgebieten). Data used for the overview map of the study region (Fig. 1) is available for free from Swisstopo and FOEN. Datasets used include: the digital height model DHM25 (https://www.swisstopo.admin.ch/de/hoehenmodell-dhm25) and the general hydrological background map (download-able via https://opendata.swiss; see https://opendata.swiss/en/dataset/generalisierte-hintergrundkarte-zur-darstellung-hydrologischer-daten).

The software used to compile the datasets are all open-source and contain the following R-packages available via CRAN: *tidyverse* (https://cran.r-project.org/web/packages/tidyverse/index.html; Wickham et al., 2019), *exactextractr* (https://cran.r-project.org/web/packages/exactextractr/index.html; Baston, 2023), *sf* (https://cran.r-project.org/web/packages/sf/index.html; Pebesma, 2018), *lfstat* (https://cran.r-project.org/web/packages/lfstat/index.html; Laaha and Koffler, 2022), *SCI* (https://cran.r-project.org/web/packages/SCI/index.html; Gudmundsson and Stagge, 2016; Stagge et al., 2015) and *stars* (https://cran.r-project.org/web/packages/stars/index.html; Pebesma and Bivand, 2023).

vand, 2023).

Available via Github are the R-packages *cwd* (Stocker (2021); available via: https://github.com/stineb/cwd), and *delayedflow* (Stoelzle et al. (2020); available via: https://modche.github.io/delayedflow/).

*Author contributions.* CNvM conceptualized the project proposal, acquired funding from the FOEN, performed the formal analysis and wrote the article. OM and BS provided guidance on the methodological aspects. CNvM, OM and BS assisted with writing the paper and

revisions.

*Competing interests.* The contact author has declared that none of the authors has any competing interests.

*Acknowledgements.* The HYD-RESPONSES project was funded by the Federal Office for the Environment (FOEN). The preliminary SPASS dataset was kindly provided by Regula Muelchi (MeteoSwiss). We thank Caroline Kan (FOEN) for her help with the catchment selection.





**Table A1.** Glossary of extracted time series variables, their description and units.

| Dataset | Variables | Variables (fullname) | Units | Producer |
|---|---|---|---|---|
| Spatial Climate Analyses | TabsD | Daily 2 m mean temperature | °C | MeteoSwiss |
| | RhiresD | Daily precipitation sums | mm | |
| | TminD | Daily 2 m minimum temperature | °C | |
| | TmaxD | Daily 2 m maximum temperature | °C | |
| | SrelD | Daily sunshine duration | % | |
| Snow Climatology for Switzerland (SPASS) | SWECLQMD | Daily snow water equivalent | mm | MeteoSwiss & SLF |
| Climatological snow data since 1998 (OSHD) | swee | Daily snow water equivalent | mm | SLF |
| | romc | Daily snowmelt-contribution to runoff | mm | |
| ERA5-Land | tp | Total precipitation | mm | ECMWF |
| | t2m | Average 2 m temperature | °C | |
| | e | Total evaporation | mm | |
| | pev | Total potential evaporation | mm | |
| | smlt | Snowmelt | mm | |
| | sd | Snow water equivalent | mm | |
| | ssr | Total solar radiation | $MJ/m^2$ | |
| | ro | Runoff | mm | |
| | sro | Surface runoff | mm | |
| | swvl1 | Soil water volume level 1 (0–7cm) | mm | |
| | swvl2 | Soil water volume level 2 (7–28cm) | mm | |
| | swvl3 | Soil water volume level 3 (28–100cm) | mm | |
| | swvl4 | Soil water volume level 4 (100–289cm) | mm | |
| Streamflow time series | Q | Daily mean streamflow | $m^3/s$ | FOEN |

# Appendix A

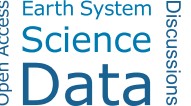

**Table A2.** Characteristics of all 184 catchments in the HYD-RESPONSES dataset (Part 1/5).

| Catchment | Water name | Place | Lon / Lat EPSG:21781 | Glaciation % | Area km² | Avg. Height m asl | Regime Type | Yearly Avg. T °C | Yearly P mm | Yearly E mm | Yearly Q mm |
|---|---|---|---|---|---|---|---|---|---|---|---|
| 0070 | Emme | Emmenmatt | 623610 / 200420 | 0.0 | 443.00 | 1065 | nivo-pluvial préalpin | 6.94 | 1539.26 | 300.29 | 856.45 |
| 0078 | Poschiavino | Le Prese | 803490 / 130520 | 4.0 | 168.00 | 2162 | nival méridional | 1.58 | 1324.78 | 175.84 | 1078.26 |
| 0155 | Emme | Wiler, Limpachmündung | 608220 / 223240 | 0.0 | 937.00 | 858 | mixed regime (>500km²) | 7.80 | 1356.58 | 313.03 | 623.66 |
| 0185 | Plessur | Chur | 757975 / 191925 | 0.0 | 264.00 | 1868 | nival alpin | 3.42 | 1179.00 | 194.41 | 915.28 |
| 0308 | Goldach | Goldach, Bleiche | 753190 / 261590 | 0.0 | 51.10 | 827 | pluvial supérieur | 8.33 | 1423.14 | 317.59 | 889.48 |
| 0352 | Linth | Linthal, Ausgleichsbecken KLL | 718285 / 197310 | 9.4 | 147.00 | 2085 | a-glacio-nival | 1.83 | 1874.45 | 169.98 | 2422.14 |
| 0403 | Inn | Cinuos-chel | 797700 / 168170 | 5.2 | 733.00 | 2456 | mixed regime (>500km²) | -0.66 | 1007.10 | 146.31 | 1007.68 |
| 0488 | Simme | Latterbach | 610680 / 167840 | 1.5 | 563.00 | 1594 | nival de transition | 4.79 | 1506.56 | 225.96 | 1108.96 |
| 0491 | Schächen | Bürglen, Galgenwäldli | 692480 / 191800 | 1.5 | 108.00 | 1728 | nivo-glaciaire | 3.55 | 1854.00 | 188.20 | 1646.44 |
| 2009 | Rhône | Porte du Scex | 557660 / 133280 | 11.0 | 5238.00 | 2127 | mixed regime (>500km²) | 1.96 | 1292.71 | 148.24 | 1116.86 |
| 2011 | Rhône | Sion | 593770 / 118630 | 14.2 | 3372.00 | 2291 | mixed regime (>500km²) | 1.00 | 1240.59 | 127.84 | 966.98 |
| 2016 | Aare | Brugg | 657000 / 259360 | 1.5 | 11681.00 | 1000 | mixed regime (>500km²) | 7.34 | 1317.32 | 291.55 | 833.95 |
| 2018 | Reuss | Mellingen | 662830 / 252580 | 1.8 | 3386.00 | 1259 | mixed regime (>500km²) | 5.98 | 1592.79 | 239.54 | 1300.70 |
| 2019 | Aare | Brienzwiler | 649930 / 177380 | 15.5 | 555.00 | 2135 | mixed regime (>500km²) | 1.41 | 1842.89 | 123.14 | 2077.41 |
| 2020 | Ticino | Bellinzona | 721245 / 117025 | 0.2 | 1517.00 | 1679 | mixed regime (>500km²) | 4.27 | 1658.40 | 209.25 | 1339.14 |
| 2024 | Rhône | Branson | 573150 / 108300 | 13.0 | 3728.00 | 2235 | mixed regime (>500km²) | 1.35 | 1249.40 | 133.83 | 1166.41 |
| 2029 | Aare | Brügg, Aegerten | 588220 / 219020 | 2.1 | 8249.00 | 1142 | mixed regime (>500km²) | 6.73 | 1366.01 | 276.35 | 908.00 |
| 2030 | Aare | Thun | 613230 / 179280 | 6.9 | 2459.00 | 1746 | mixed regime (>500km²) | 3.72 | 1604.05 | 176.90 | 1438.59 |
| 2033 | Vorderrhein | Ilanz | 735000 / 182030 | 1.8 | 774.00 | 2030 | mixed regime (>500km²) | 2.28 | 1534.35 | 157.08 | 1340.76 |
| 2034 | Broye | Payerne, Caserne d'aviation | 561660 / 187320 | 0.0 | 416.00 | 715 | pluvial inférieur | 9.19 | 1186.40 | 322.34 | 564.59 |
| 2044 | Thur | Andelfingen | 693510 / 272500 | 0.0 | 1702.00 | 770 | mixed regime (>500km²) | 8.25 | 1392.81 | 333.74 | 857.45 |
| 2053 | Drance | Martigny, Pont de Rossettan | 570930 / 105200 | 11.3 | 676.00 | 2250 | mixed regime (>500km²) | 1.40 | 1269.62 | 138.70 | 462.70 |
| 2056 | Reuss | Seedorf | 690085 / 193210 | 6.4 | 833.00 | 2013 | mixed regime (>500km²) | 1.96 | 1681.87 | 150.85 | 1624.38 |
| 2063 | Aare | Murgenthal | 629530 / 235090 | 1.7 | 10059.00 | 1066 | mixed regime (>500km²) | 7.04 | 1346.17 | 284.49 | 888.26 |
| 2070 | Emme | Emmenmatt, nur Hauptstation | 623610 / 200420 | 0.0 | 443.00 | 1065 | nivo-pluvial préalpin | 6.94 | 1539.26 | 300.29 | 835.36 |
| 2078 | Poschiavino | Le Prese, stazione principale | 803490 / 130520 | 4.0 | 168.00 | 2162 | nival méridional | 1.58 | 1324.78 | 175.84 | 1064.96 |
| 2084 | Muota | Ingenbohl | 688230 / 206140 | 0.0 | 317.00 | 1363 | nival de transition | 5.45 | 1958.67 | 237.00 | 1915.23 |
| 2085 | Aare | Hagneck | 580680 / 211650 | 3.4 | 5112.00 | 1368 | mixed regime (>500km²) | 5.61 | 1452.15 | 237.22 | 1068.53 |
| 2086 | Brenno | Loderio | 717770 / 137270 | 0.3 | 400.00 | 1815 | nival méridional | 3.68 | 1618.84 | 185.01 | 342.59 |
| 2087 | Reuss | Andermatt | 688120 / 166320 | 2.9 | 190.00 | 2284 | b-glacio-nival | 0.59 | 1709.03 | 116.01 | 1167.49 |
| 2091 | Rhein | Rheinfelden, Messstation | 627190 / 267840 | 0.8 | 34524.00 | 1068 | mixed regime (>500km²) | 6.68 | 1351.54 | 281.92 | 935.92 |
| 2099 | Limmat | Zürich, Unterhard | 682055 / 249430 | 0.8 | 2174.00 | 1194 | mixed regime (>500km²) | 6.28 | 1719.03 | 264.98 | 1353.24 |
| 2102 | Sarner Aa | Sarnen | 661460 / 194220 | 0.0 | 269.00 | 1281 | downstream lake | 5.99 | 1648.90 | 224.99 | 1167.84 |
| 2104 | Linth | Weesen, Biäsche | 725160 / 221380 | 1.6 | 1062.00 | 1584 | mixed regime (>500km²) | 4.39 | 1785.64 | 221.45 | 1538.41 |
| 2105 | Inn | St. Moritzbad | 783910 / 150960 | 3.8 | 155.00 | 2399 | b-glacio-nival | -0.33 | 1055.10 | 161.39 | 1145.54 |
| 2106 | Birs | Münchenstein, Hofmatt | 613570 / 263080 | 0.0 | 887.00 | 728 | mixed regime (>500km²) | 8.53 | 1206.82 | 335.73 | 545.50 |
| 2109 | Lütschine | Gsteig | 633130 / 168200 | 13.5 | 381.00 | 2050 | a-glacio-nival | 2.11 | 1780.73 | 119.63 | 1580.50 |
| 2110 | Reuss | Mühlau, Hünenberg | 672520 / 230600 | 2.2 | 2902.00 | 1371 | mixed regime (>500km²) | 5.42 | 1641.78 | 226.17 | 1399.39 |
| 2112 | Sitter | Appenzell | 749040 / 244220 | 0.1 | 74.40 | 1256 | nival de transition | 6.22 | 1896.65 | 345.94 | 1421.58 |
| 2117 | Drance de Bagnes | Le Châble, Villette | 582550 / 103270 | 22.1 | 254.00 | 2609 | b-glaciaire | -0.59 | 1274.15 | 118.82 | 254.59 |
| 2119 | Sarine | Fribourg | 579420 / 183670 | 0.2 | 1271.00 | 1247 | mixed regime (>500km²) | 6.35 | 1420.49 | 276.29 | 975.93 |
| 2122 | Birse | Moutier, La Charrue | 595740 / 237010 | 0.0 | 186.00 | 921 | nivo-pluvial jurassien | 7.49 | 1371.76 | 333.62 | 520.67 |

T = Temperature, P = Precipitation, E = Evaporation, Q = Streamflow/Runoff



**Table A3.** Characteristics of all 184 catchments in the HYD-RESPONSES dataset (Part 2/5).

| Catchment | Water name | Place | Lon / Lat EPSG:21781 | Glaciation % | Area km² | Avg. Height m asl | Regime Type | Yearly Avg. T °C | Yearly P mm | Yearly E mm | Yearly Q mm |
|---|---|---|---|---|---|---|---|---|---|---|---|
| 2125 | Lorze | Frauenthal | 674715 / 229845 | 0.0 | 262.00 | 678 | downstream lake | 8.91 | 1427.56 | 309.43 | 911.18 |
| 2126 | Murg | Wängi | 714105 / 261720 | 0.0 | 80.20 | 652 | pluvial inférieur | 8.72 | 1282.82 | 340.64 | 693.21 |
| 2132 | Töss | Neftenbach | 691460 / 263820 | 0.0 | 343.00 | 658 | pluvial inférieur | 8.89 | 1331.52 | 339.46 | 701.10 |
| 2135 | Aare | Bern, Schönau | 600710 / 198000 | 5.8 | 2941.00 | 1596 | mixed regime (>500km²) | 4.43 | 1542.51 | 196.01 | 1317.74 |
| 2139 | Rheintaler Binnenkanal | St. Margrethen | 767160 / 257780 | 0.0 | 175.00 | 710 | artificial waterbody | 9.01 | 1451.16 | 310.08 | 2038.65 |
| 2141 | Albula | Tiefencastel | 763420 / 170145 | 0.5 | 529.00 | 2128 | mixed regime (>500km²) | 1.53 | 1018.61 | 159.38 | 904.27 |
| 2143 | Rhein | Rekingen | 667060 / 269230 | 0.2 | 14767.00 | 1131 | mixed regime (>500km²) | 5.83 | 1296.20 | 276.20 | 945.75 |
| 2150 | Landquart | Felsenbach | 765365 / 204910 | 0.7 | 614.00 | 1797 | mixed regime (>500km²) | 3.34 | 1289.45 | 188.82 | 1203.19 |
| 2151 | Simme | Oberwil | 600060 / 167090 | 2.4 | 344.00 | 1641 | nival de transition | 4.52 | 1536.17 | 215.10 | 1084.54 |
| 2152 | Reuss | Luzern, Geissmattbrücke | 665330 / 211800 | 2.8 | 2254.00 | 1504 | mixed regime (>500km²) | 4.72 | 1683.08 | 207.59 | 1526.87 |
| 2155 | Emme | Wiler, Limpachmündung, nur Hauptstation | 608220 / 223240 | 0.0 | 924.00 | 863 | mixed regime (>500km²) | 7.80 | 1356.58 | 313.03 | 316.15 |
| 2159 | Gürbe | Belp, Mülimatt | 604810 / 192680 | 0.0 | 116.00 | 846 | pluvial supérieur | 8.05 | 1236.50 | 298.87 | 715.53 |
| 2160 | Sarine | Broc, Château d'en bas | 573520 / 161345 | 0.3 | 636.00 | 1500 | mixed regime (>500km²) | 5.12 | 1500.63 | 248.29 | 1014.01 |
| 2161 | Massa | Blatten bei Naters | 643700 / 137290 | 56.5 | 196.00 | 2937 | a-glaciaire | -2.89 | 2036.03 | 45.98 | 2433.50 |
| 2167 | Tresa | Ponte Tresa, Rocchetta | 709580 / 92145 | 0.0 | 609.00 | 803 | mixed regime (>500km²) | 9.59 | 1789.28 | 351.80 | 1107.04 |
| 2170 | Arve | Genève, Bout du Monde | 501220 / 115120 | 5.1 | 1973.00 | 1370 | mixed regime (>500km²) | 5.65 | 1505.42 | 249.36 | 1153.49 |
| 2174 | Rhône | Chancy, Aux Ripes | 486600 / 112340 | 6.6 | 10308.00 | 1569 | mixed regime (>500km²) | 4.32 | 1324.91 | 216.70 | 1027.76 |
| 2176 | Sihl | Zürich, Sihlhölzli | 682145 / 246890 | 0.0 | 343.00 | 1045 | nivo-pluvial préalpin | 6.97 | 1787.58 | 294.48 | 621.39 |
| 2179 | Sense | Thörishaus, Sensematt | 593350 / 193020 | 0.0 | 351.00 | 1071 | nivo-pluvial préalpin | 7.22 | 1404.55 | 306.68 | 756.67 |
| 2181 | Thur | Halden | 733560 / 263180 | 0.0 | 1085.00 | 908 | mixed regime (>500km²) | 7.61 | 1585.63 | 329.74 | 1082.92 |
| 2185 | Plessur | Chur, nur Hauptstation | 757975 / 191925 | 0.0 | 264.00 | 1868 | nival alpin | 3.42 | 1179.00 | 194.41 | 693.27 |
| 2187 | Werdenberger Binnenkanal | Salez | 756795 / 234005 | 0.0 | 183.00 | 1003 | artificial waterbody | 7.74 | 1547.48 | 279.64 | 1344.65 |
| 2199 | Wiese | Basel | 611800 / 269700 | 0.0 | 442.00 | 720 | pluvial jurassien | 10.67 | 1508.42 | 342.75 | 800.00 |
| 2200 | Weisse Lütschine | Zweilütschinen | 635310 / 164550 | 13.1 | 165.00 | 2165 | a-glacio-nival | 1.58 | 1767.24 | 112.23 | 1531.27 |
| 2202 | Ergolz | Liestal | 622270 / 259950 | 0.0 | 261.00 | 588 | pluvial jurassien | 9.55 | 1076.57 | 341.74 | 436.87 |
| 2203 | Grande Eau | Aigle | 563975 / 129825 | 0.8 | 132.00 | 1562 | nival de transition | 4.96 | 1617.15 | 240.72 | 1082.21 |
| 2205 | Aare | Untersiggenthal, Stilli | 659970 / 263180 | 1.4 | 17553.00 | 1064 | mixed regime (>500km²) | 6.99 | 1416.13 | 279.19 | 984.66 |
| 2206 | Melera | Melera (Valle Morobbia) | 726988 / 114670 | 0.0 | 1.07 | 1423 | nivo-pluvial méridional | 5.88 | 1712.31 | 290.07 | 1297523.71 |
| 2210 | Doubs | Ocourt | 572530 / 244460 | 0.0 | 1275.00 | 952 | mixed regime (>500km²) | 7.10 | 1499.13 | 346.76 | 790.06 |
| 2215 | Saane | Laupen | 584440 / 195300 | 0.1 | 1862.00 | 1137 | mixed regime (>500km²) | 6.87 | 1373.00 | 288.00 | 866.85 |
| 2219 | Simme | Oberried / Lenk | 602630 / 141660 | 22.6 | 34.80 | 2347 | b-glaciaire | 0.92 | 1779.85 | 171.95 | 2130.36 |
| 2232 | Allenbach | Adelboden | 608710 / 148300 | 0.0 | 28.80 | 1863 | nival alpin | 3.64 | 1557.08 | 174.36 | 1332.39 |
| 2239 | Spöl | Punt dal Gall | 811020 / 167920 | 0.3 | 295.00 | 2389 | nivo-glaciaire | -0.61 | 940.86 | 152.75 | 105.09 |
| 2243 | Limmat | Baden, Limmatpromenade | 665640 / 258690 | 0.7 | 2394.00 | 1131 | mixed regime (>500km²) | 6.59 | 1662.82 | 272.41 | 1311.73 |
| 2244 | Krummbach | Klusmatten | 644500 / 119420 | 0.4 | 19.40 | 2271 | nival méridional | 1.35 | 1342.63 | 166.60 | 1232.18 |
| 2247 | Doubs | Sortie du lac des Brenets | 544560 / 214880 | 0.0 | 867.00 | 977 | nivo-glaciaire | 6.41 | 1011.37 | 350.16 | 635.02 |
| 2251 | Rotenbach | Plaffeien, Schwyberg | 587980 / 170590 | 0.0 | 1.69 | 1455 | nival de transition | 5.70 | 1688.19 | 296.97 | 1427822.37 |
| 2252 | Schwändlibach | Plaffeien, Schwyberg | 588340 / 171015 | 0.0 | 1.38 | 1439 | nival de transition | 5.76 | 1662.14 | 296.97 | 832954.23 |
| 2256 | Rosegbach | Pontresina | 788810 / 151690 | 21.7 | 66.50 | 2704 | a-glaciaire | -1.75 | 1137.41 | 119.24 | 1398.90 |
| 2262 | Berninabach | Pontresina | 789440 / 151320 | 14.4 | 107.00 | 2615 | a-glacio-nival | -1.18 | 1203.87 | 131.81 | 1411.84 |
| 2263 | Chamuerabach | La Punt-Chamues-ch | 791430 / 160600 | 0.1 | 73.40 | 2548 | nivo-glaciaire | -1.10 | 1011.37 | 145.82 | 930.92 |
| 2265 | Inn | Tarasp | 816800 / 185910 | 3.0 | 1581.00 | 2384 | mixed regime (>500km²) | -0.37 | 992.18 | 147.59 | 383.43 |
| 2268 | Rhone | Gletsch | 670810 / 157200 | 41.8 | 39.40 | 2710 | a-glaciaire | -1.75 | 1937.62 | 75.90 | 2342.03 |
| 2269 | Lonza | Blatten | 629130 / 140910 | 24.7 | 77.40 | 2624 | a-glaciaire | -1.28 | 1566.54 | 86.75 | 1924.12 |
| 2276 | Grosstalbach | Isenthal | 685500 / 196050 | 6.7 | 43.90 | 1819 | nival alpin | 3.28 | 1731.55 | 227.73 | 1270.99 |

T = Temperature, P = Precipitation, E = Evaporation, Q = Streamflow/Runoff





**Table A4.** Characteristics of all 184 catchments in the HYD-RESPONSES dataset (Part 3/5).

| Catchment | Water name | Place | Lon / Lat EPSG:21781 | Glaciation % | Area km² | Avg. Height m asl | Regime Type | Yearly Avg. T °C | Yearly P mm | Yearly E mm | Yearly Q mm |
|---|---|---|---|---|---|---|---|---|---|---|---|
| 2282 | Sperbelgraben | Wasen, Kurzeneialp | 630725 / 207270 | 0.0 | 0.56 | 1070 | nivo-pluvial préalpin | 7.06 | 1631.78 | 342.53 | 883404.51 |
| 2283 | Rappengraben | Wasen, Riedbad | 634340 / 207350 | 0.0 | 0.60 | 1142 | nivo-pluvial préalpin | 6.87 | 1656.81 | 355.40 | 1064734.56 |
| 2288 | Rhein | Neuhausen, Flurlingerbrücke | 689145 / 281975 | 0.3 | 11930.00 | 1239 | mixed regime (>500km²) | 4.59 | 1295.50 | 261.93 | 960.66 |
| 2289 | Rhein | Basel, Rheinhalle | 613400 / 267650 | 0.8 | 35878.00 | 1052 | mixed regime (>500km²) | 6.78 | 1343.83 | 284.04 | 919.47 |
| 2290 | Areuse | St-Sulpice | 532980 / 195880 | 0.0 | 104.00 | 1110 | nivo-pluvial jurassien | 5.67 | 1500.18 | 344.81 | 1408.21 |
| 2299 | Alpbach | Erstfeld, Bodenberg | 688560 / 185120 | 19.7 | 20.70 | 2205 | b-glaciaire | 1.07 | 1669.66 | 171.48 | 2406.18 |
| 2300 | Minster | Euthal, Rüti | 704425 / 215310 | 0.0 | 59.10 | 1352 | nival de transition | 5.53 | 2115.46 | 259.83 | 1639.61 |
| 2303 | Thur | Jonschwil, Mühlau | 723675 / 252720 | 0.0 | 493.00 | 1021 | nivo-pluvial préalpin | 6.93 | 1757.19 | 320.59 | 1285.82 |
| 2304 | Ova dal Fuorn | Zernez, Punt la Drossa | 810560 / 170790 | 0.0 | 55.30 | 2327 | nival alpin | -0.46 | 937.69 | 150.89 | 586.32 |
| 2305 | Glatt | Herisau, Zellersmühle | 737270 / 251290 | 0.0 | 16.70 | 829 | pluvial supérieur | 8.18 | 1491.95 | 329.49 | 1063.60 |
| 2307 | Suze | Sonceboz | 579810 / 227350 | 0.0 | 127.00 | 1036 | nivo-pluvial jurassien | 6.97 | 1332.88 | 340.21 | 1008.10 |
| 2308 | Goldach | Goldach, Bleiche, nur Hauptstation | 753190 / 261590 | 0.0 | 50.40 | 832 | pluvial supérieur | 8.33 | 1423.14 | 317.59 | 853.11 |
| 2312 | Aach | Salmsach, Hungerbühl | 744110 / 268400 | 0.0 | 47.40 | 467 | pluvial inférieur | 9.68 | 1019.41 | 335.09 | 488.00 |
| 2319 | Ova da Cluozza | Zernez | 804930 / 174830 | 0.0 | 27.00 | 2371 | nivo-glaciaire | -0.47 | 919.61 | 150.80 | 888.70 |
| 2321 | Cassarate | Pregassona | 718010 / 97380 | 0.0 | 75.80 | 987 | pluvio-nival méridional | 8.52 | 1900.06 | 330.75 | 983.33 |
| 2327 | Dischmabach | Davos, Kriegsmatte | 786220 / 183370 | 0.7 | 42.90 | 2376 | b-glacio-nival | 0.15 | 1015.77 | 147.17 | 1242.21 |
| 2342 | Saltina | Brig | 642220 / 129630 | 2.5 | 76.50 | 2014 | nivo-glaciaire | 2.53 | 1165.38 | 167.60 | 948.67 |
| 2343 | Langete | Huttwil, Häberenbad | 629560 / 219135 | 0.0 | 59.90 | 760 | pluvial inférieur | 8.14 | 1276.02 | 329.38 | 618.66 |
| 2346 | Rhone | Brig | 641340 / 129700 | 19.2 | 906.00 | 2339 | mixed regime (>500km²) | 0.31 | 1630.34 | 103.68 | 1481.12 |
| 2347 | Riale di Roggiasca | Roveredo, Bacino di compenso | 733545 / 118160 | 0.0 | 8.12 | 1702 | nivo-pluvial méridional | 4.11 | 1684.56 | 288.12 | 1869.20 |
| 2349 | Breggia | Chiasso, Ponte di Polenta | 722315 / 78320 | 0.0 | 47.10 | 933 | pluvio-nival méridional | 8.58 | 1726.83 | 382.64 | 712.61 |
| 2351 | Vispa | Visp | 634050 / 125900 | 23.1 | 786.00 | 2648 | mixed regime (>500km²) | -0.92 | 1125.42 | 116.03 | 684.02 |
| 2352 | Linth | Linthal, Ausgleichsbecken KLL, nur Haupt | 718285 / 197310 | 9.4 | 147.00 | 2085 | a-glacio-nival | 1.83 | 1874.45 | 169.98 | 925.43 |
| 2355 | Landwasser | Davos, Frauenkirch | 779640 / 181200 | 0.3 | 184.00 | 2224 | nivo-glaciaire | 0.97 | 1063.29 | 151.03 | 926.33 |
| 2356 | Riale di Calneggia | Cavergno, Pontit | 684970 / 135960 | 0.0 | 23.90 | 2003 | nival méridional | 3.08 | 1868.56 | 188.46 | 1922.63 |
| 2364 | Ticino | Piotta | 694610 / 152450 | 0.3 | 159.00 | 2071 | nival méridional | 2.16 | 1803.44 | 136.09 | 413.29 |
| 2366 | Poschiavino | La Rösa | 802120 / 142010 | 0.0 | 14.10 | 2285 | nival méridional | 0.86 | 1398.05 | 162.22 | 1189.26 |
| 2368 | Maggia | Locarno, Solduno | 703100 / 113860 | 0.3 | 927.00 | 1530 | mixed regime (>500km²) | 5.63 | 1946.42 | 245.77 | 783.83 |
| 2369 | Mentue | Yvonand, La Mauguettaz | 545440 / 180875 | 0.0 | 105.00 | 675 | pluvial jurassien | 9.35 | 1081.13 | 328.32 | 457.37 |
| 2370 | Doubs | Le Noirmont, La Goule | 561430 / 231050 | 0.0 | 1047.00 | 977 | mixed regime (>500km²) | 6.69 | 1534.75 | 348.30 | 795.29 |
| 2371 | Orbe | Le Chenit, Frontière | 501445 / 156305 | 0.0 | 45.90 | 1235 | nivo-pluvial jurassien | 6.42 | 1901.34 | 337.96 | 615.79 |
| 2372 | Linth | Mollis, Linthbrücke | 723985 / 217965 | 2.9 | 600.00 | 1743 | mixed regime (>500km²) | 3.49 | 1848.95 | 197.67 | 1687.89 |
| 2374 | Necker | Mogelsberg, Aachsäge | 727110 / 247290 | 0.0 | 88.10 | 956 | nivo-pluvial préalpin | 7.27 | 1718.29 | 338.15 | 1142.38 |
| 2378 | Orbe | Orbe, Le Chalet | 530080 / 175560 | 0.0 | 343.00 | 1139 | nivo-pluvial jurassien | 6.73 | 1692.73 | 341.81 | 1016.62 |
| 2386 | Murg | Frauenfeld | 709540 / 269660 | 0.0 | 213.00 | 597 | pluvial inférieur | 8.98 | 1178.35 | 343.21 | 567.47 |
| 2387 | Hinterrhein | Fürstenau | 753570 / 175730 | 0.6 | 1577.00 | 2127 | mixed regime (>500km²) | 1.47 | 1147.93 | 165.74 | 767.37 |
| 2403 | Inn | Cinuos-chel, nur Hauptstation | 797700 / 168170 | 5.2 | 733.00 | 2456 | mixed regime (>500km²) | -0.66 | 1007.10 | 146.31 | 212.71 |
| 2409 | Emme | Eggiwil, Heidbiel | 627910 / 191180 | 0.0 | 124.00 | 1281 | nivo-pluvial préalpin | 6.10 | 1604.24 | 270.21 | 1061.40 |
| 2410 | Liechtensteiner Binnenkanal | Ruggell | 757750 / 234590 | 0.0 | 116.00 | 853 | artificial waterbody | 8.58 | 1286.29 | 264.96 | 1321.13 |
| 2412 | Sionge | Vuippens, Château | 572420 / 167540 | 0.0 | 43.40 | 865 | nivo-pluvial préalpin | 8.10 | 1298.00 | 302.73 | 802.84 |
| 2414 | Rietholzbach | Mosnang, Rietholz | 718840 / 248440 | 0.0 | 3.19 | 794 | pluvial supérieur | 8.13 | 1476.78 | 336.69 | 1006909.39 |
| 2415 | Glatt | Rheinsfelden | 678040 / 269720 | 0.0 | 417.00 | 503 | downstream lake | 9.70 | 1165.36 | 340.63 | 590.10 |
| 2416 | Aabach | Hitzkirch, Richensee | 661390 / 230220 | 0.0 | 73.30 | 581 | downstream lake | 9.46 | 1163.00 | 317.37 | 535.90 |
| 2417 | Suhre | Oberkirch | 651320 / 223140 | 0.0 | 75.60 | 583 | downstream lake | 9.39 | 1139.68 | 312.72 | 510.78 |
| 2418 | Julia | Tiefencastel | 763570 / 169910 | 0.2 | 325.00 | 2196 | nivo-glaciaire | 1.10 | 1058.77 | 161.39 | 97.09 |
| 2419 | Rhone | Reckingen | 661910 / 146780 | 11.8 | 214.00 | 2305 | a-glacio-nival | 0.30 | 1814.00 | 105.51 | 1424.61 |
| 2420 | Moesa | Lumino, Sassello | 724765 / 120360 | 0.1 | 472.00 | 1667 | nivo-pluvial méridional | 3.98 | 1619.63 | 234.73 | 1339.42 |

T = Temperature, P = Precipitation, E = Evaporation, Q = Streamflow/Runoff





**Table A5.** Characteristics of all 184 catchments in the HYD-RESPONSES dataset (Part 4/5).

| Catchment | Water name | Place | Lon / Lat EPSG:21781 | Glaciation % | Area km² | Avg. Height m asl | Regime Type | Yearly Avg. T °C | Yearly P mm | Yearly E mm | Yearly Q mm |
|---|---|---|---|---|---|---|---|---|---|---|---|
| 2426 | Seez | Mels | 750410 / 212510 | 0.1 | 106.00 | 1803 | nival alpin | 3.55 | 1578.39 | 217.42 | 640.88 |
| 2430 | Rein da Sumvig | Sumvitg, Encardens | 718810 / 167690 | 1.7 | 21.80 | 2457 | b-glacio-nival | -0.15 | 1581.92 | 160.82 | 2183.90 |
| 2432 | Venoge | Ecublens, Les Bois | 532040 / 154160 | 0.0 | 228.00 | 686 | nivo-pluvial jurassien | 9.62 | 1148.17 | 332.99 | 539.44 |
| 2433 | Aubonne | Allaman, Le Coulet | 520720 / 147410 | 0.0 | 105.00 | 952 | nivo-pluvial jurassien | 8.21 | 1444.62 | 340.37 | 1587.98 |
| 2434 | Dünnern | Olten, Hammermühle | 634330 / 244480 | 0.0 | 234.00 | 711 | pluvial jurassien | 8.50 | 1210.83 | 334.79 | 437.11 |
| 2436 | Chli Schliere | Alpnach, Chilch Erli | 663800 / 199570 | 0.0 | 21.60 | 1345 | nivo-pluvial préalpin | 5.96 | 1876.39 | 271.62 | 966.44 |
| 2437 | Parimbot | Ecublens, Eschiens | 552060 / 161650 | 0.0 | 6.92 | 716 | pluvial jurassien | 9.50 | 1182.10 | 311.77 | 717260.34 |
| 2450 | Wigger | Zofingen | 637580 / 237080 | 0.0 | 366.00 | 656 | pluvial inférieur | 8.80 | 1182.26 | 328.47 | 461.60 |
| 2457 | Aare | Ringgenberg, Goldswil | 633730 / 171510 | 12.1 | 1138.00 | 1951 | mixed regime (>500km²) | 2.47 | 1761.77 | 138.78 | 1715.63 |
| 2458 | Seyon | Valangin | 559370 / 206810 | 0.0 | 112.00 | 978 | nivo-pluvial jurassien | 7.54 | 1292.29 | 350.00 | 214.91 |
| 2461 | Magliasina | Magliaso, Ponte | 711620 / 93290 | 0.0 | 34.40 | 926 | pluvio-nival méridional | 8.91 | 1938.53 | 357.01 | 1117.90 |
| 2468 | Sitter | St. Gallen, Bruggen/Au | 742540 / 253230 | 0.0 | 261.00 | 1042 | nivo-pluvial préalpin | 7.22 | 1722.67 | 343.20 | 1208.91 |
| 2471 | Murg | Murgenthal, Walliswil | 629340 / 233555 | 0.0 | 183.00 | 653 | pluvial inférieur | 8.60 | 1191.59 | 333.34 | 556.21 |
| 2473 | Rhein | Diepoldsau, Rietbrücke | 766280 / 250360 | 0.6 | 6299.00 | 1771 | mixed regime (>500km²) | 3.17 | 1327.19 | 193.52 | 1163.69 |
| 2474 | Calancasca | Buseno | 729440 / 127180 | 0.2 | 121.00 | 1931 | nival méridional | 2.65 | 1673.78 | 227.69 | 1113.07 |
| 2475 | Maggia | Bignasco, Ponte nuovo | 690040 / 132550 | 0.9 | 316.00 | 1879 | nival méridional | 3.67 | 1939.62 | 187.23 | 415.67 |
| 2477 | Lorze | Zug, Letzi | 680600 / 226070 | 0.0 | 100.00 | 818 | downstream lake | 8.15 | 1560.20 | 295.43 | 925.59 |
| 2478 | Birse | Soyhières, Bois du Treuil | 596780 / 249070 | 0.0 | 569.00 | 805 | nivo-pluvial jurassien | 8.06 | 1265.37 | 334.76 | 580.60 |
| 2480 | Areuse | Boudry | 554350 / 199940 | 0.0 | 378.00 | 1077 | nivo-pluvial jurassien | 6.27 | 1464.77 | 347.62 | 906.66 |
| 2481 | Engelberger Aa | Buochs, Flugplatz | 673555 / 202870 | 2.5 | 228.00 | 1609 | b-glacio-nival | 4.30 | 1693.58 | 196.94 | 1705.73 |
| 2485 | Allaine | Boncourt, Frontière | 567830 / 261200 | 0.0 | 212.00 | 562 | pluvial jurassien | 9.54 | 1108.82 | 343.10 | 464.95 |
| 2486 | Veveyse | Vevey, Copet | 554675 / 146565 | 0.0 | 64.50 | 1098 | nivo-pluvial préalpin | 7.37 | 1497.99 | 307.89 | 955.88 |
| 2487 | Kleine Emme | Werthenstein, Chappelboden | 647870 / 209510 | 0.0 | 311.00 | 1167 | nivo-pluvial préalpin | 6.61 | 1695.60 | 279.00 | 1095.28 |
| 2488 | Simme | Latterbach | 610680 / 167840 | 1.5 | 563.00 | 1594 | nival de transition | 4.79 | 1506.56 | 225.96 | 342.76 |
| 2490 | Allondon | Dardagny, Les Granges | 488880 / 119460 | 0.0 | 119.00 | 760 | nivo-pluvial jurassien | 10.64 | 1372.60 | 326.46 | 854.70 |
| 2491 | Schächen | Bürglen, Galgenwäldli, nur Hauptstation | 692480 / 191800 | 1.5 | 108.00 | 1728 | nivo-glaciaire | 3.55 | 1854.00 | 188.20 | 1397.24 |
| 2493 | Promenthouse | Gland, Route Suisse | 510080 / 140080 | 0.0 | 120.00 | 1027 | nivo-pluvial jurassien | 7.73 | 1577.83 | 338.92 | 430.80 |
| 2494 | Ticino | Pollegio, Campagna | 716120 / 135330 | 0.2 | 444.00 | 1796 | nival méridional | 3.85 | 1710.59 | 173.72 | 1438.77 |
| 2497 | Luthern | Nebikon | 640560 / 226740 | 0.0 | 105.00 | 749 | pluvial inférieur | 8.31 | 1268.69 | 334.87 | 429.99 |
| 2498 | Glenner | Castrisch | 735330 / 181790 | 1.1 | 381.00 | 2022 | nivo-glaciaire | 2.11 | 1307.02 | 168.97 | 734.66 |
| 2500 | Worble | Ittigen | 603005 / 202455 | 0.0 | 67.10 | 666 | pluvial inférieur | 8.72 | 1174.24 | 317.69 | 475.10 |
| 2602 | Rhein | Doma/Ems | 753890 / 189370 | 0.9 | 3229.00 | 2013 | mixed regime (>500km²) | 2.19 | 1277.76 | 168.28 | 1122.84 |
| 2603 | Ilfis | Langnau | 627320 / 198600 | 0.0 | 187.00 | 1039 | nivo-pluvial préalpin | 7.05 | 1619.21 | 318.70 | 882.14 |
| 2604 | Biber | Biberbrugg | 697240 / 223280 | 0.0 | 31.90 | 1003 | nivo-pluvial préalpin | 7.00 | 1789.15 | 287.37 | 1085.77 |
| 2605 | Verzasca | Lavertezzo, Campiöi | 708420 / 122920 | 0.0 | 185.00 | 1651 | nivo-pluvial méridional | 5.11 | 2013.18 | 260.43 | 1846.37 |
| 2606 | Rhône | Genève, Halle de l'Ile | 499890 / 117850 | 7.2 | 8000.00 | 1658 | mixed regime (>500km²) | 4.08 | 1286.36 | 204.12 | 996.79 |
| 2607 | Goneri | Oberwald | 670467 / 153932 | 4.0 | 38.50 | 2383 | b-glacio-nival | 0.07 | 1976.16 | 121.82 | 2011.50 |
| 2608 | Sellenbodenbach | Neuenkirch | 658530 / 218290 | 0.0 | 10.40 | 608 | pluvial inférieur | 9.33 | 1193.97 | 305.25 | 637.12 |

T = Temperature, P = Precipitation, E = Evaporation, Q = Streamflow/Runoff





**Table A6.** Characteristics of all 184 catchments in the HYD-RESPONSES dataset (Part 5/5).

| Catchment | Water name | Place | Lon / Lat EPSG:21781 | Glaciation % | Area km² | Avg. Height m asl | Regime Type | Yearly Avg. T °C | Yearly P mm | Yearly E mm | Yearly Q mm |
|---|---|---|---|---|---|---|---|---|---|---|---|
| 2609 | Alp | Einsiedeln | 698640 / 223020 | 0.0 | 46.70 | 1157 | nivo-pluvial préalpin | 6.35 | 1939.89 | 279.80 | 1459.07 |
| 2610 | Scheulte | Vicques | 599485 / 244150 | 0.0 | 72.70 | 792 | nivo-pluvial jurassien | 8.20 | 1260.81 | 333.18 | 635.64 |
| 2612 | Riale di Pincascia | Lavertezzo | 708060 / 123950 | 0.0 | 44.50 | 1705 | nivo-pluvial méridional | 4.89 | 1978.57 | 277.65 | 1911.93 |
| 2617 | Rom | Müstair | 830800 / 168700 | 0.0 | 128.00 | 2184 | nival alpin | 1.03 | 844.68 | 158.63 | 577.25 |
| 2620 | Mera | Soglio | 760770 / 133450 | 7.4 | 177.00 | 2173 | b-glacio-nival | 0.95 | 1333.42 | 191.09 | 361.81 |
| 2629 | Vedeggio | Agno, stazione principale | 714110 / 95680 | 0.0 | 99.90 | 921 | pluvio-nival méridional | 8.87 | 1904.96 | 334.27 | 656.21 |
| 2630 | Sionne | Sion | 594400 / 119900 | 0.0 | 27.60 | 1577 | nival alpin | 5.35 | 1355.03 | 186.55 | 224.58 |
| 2631 | Hinterrhein | Hinterrhein, Schiessplatz | 733706 / 153945 | 9.1 | 41.50 | 2430 | a-glacio-nival | -0.38 | 1704.08 | 164.40 | 799.71 |
| 2634 | Kleine Emme | Emmen | 663700 / 213630 | 0.0 | 478.00 | 1054 | nivo-pluvial préalpin | 7.15 | 1610.61 | 284.17 | 994.18 |
| 2635 | Grossbach | Einsiedeln, Gross | 700710 / 218125 | 0.0 | 8.95 | 1283 | nivo-pluvial préalpin | 5.90 | 1952.69 | 299.45 | 1387.71 |
| 2640 | Sorne | Delémont, Pré-Guillaume | 593380 / 245940 | 0.0 | 214.00 | 779 | nivo-pluvial jurassien | 8.20 | 1233.60 | 335.80 | 603.72 |
| 2646 | Kander | Emdthal | 617790 / 168400 | 5.1 | 487.00 | 1860 | b-glacio-nival | 3.38 | 1486.71 | 167.85 | 1305.44 |

T = Temperature, P = Precipitation, E = Evaporation, Q = Streamflow/Runoff



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



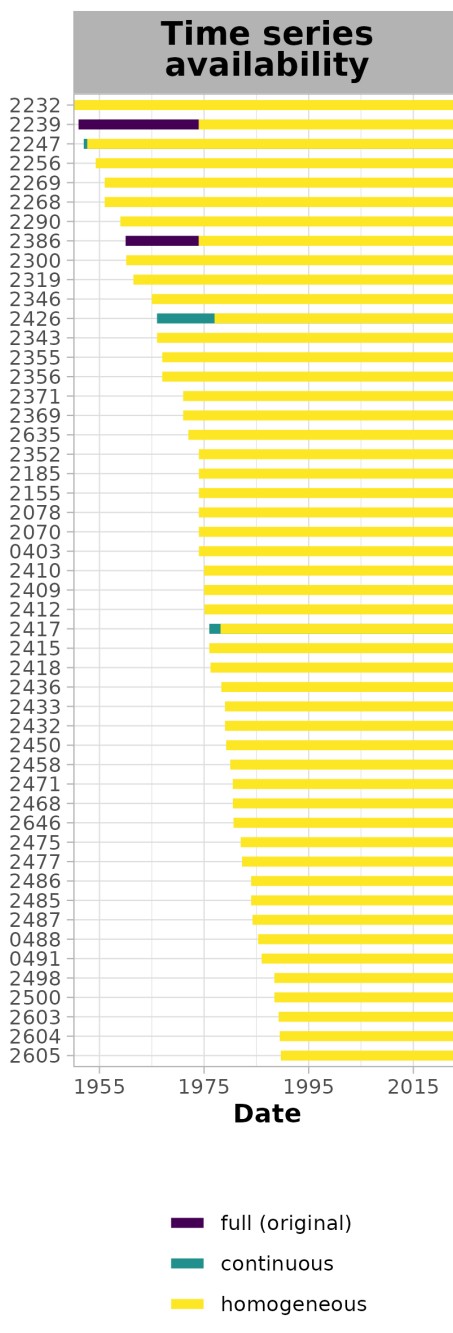

**Figure 4.** Streamflow time series availability for 50 example catchments. The colours indicate the periods covered by availability type. Complete is equivalent to the original time series provided by the FOEN. Continuous denotes the gap-checked time series and the homogeneous period accounts for homogeneity (starting at a breakpoint). In the case of overlapping periods, only the most important period type for analysis (e.g., homogeneous) is displayed. The importance of the periods for analysis is defined as follows: *homogeneous* is more important than *continuous* is more important than *full (original)*.





**Figure 5.** Evaluation statistics for the transformation of standardized (drought) indices. Information on the normality tests (*p*-values), flags and implausible/missing values for four example candidate distributions for the Standardized Precipitation and Evaporation Index (SPEI; Vicente-Serrano et al., 2010). The circle size indicates the number of missing and implausible values Colours show the number of flags (= convergence issues) returned by the fitting function of the *SCI* R-package (Stagge et al., 2015; Gudmundsson and Stagge, 2016) for all days of the year (DOY). The maximum number of flags is equivalent to 366. Median *p*-values of the Shapiro-Wilks normality test (Shapiro and Wilk, 1965) were calculated by considering all catchments and are coloured in red in case of rejection ($p < 0.05$). The final HYD-RESPONSES dataset only provides SPEIs fitted by the *genlog*-distribution (best choice based on the evaluation criteria).



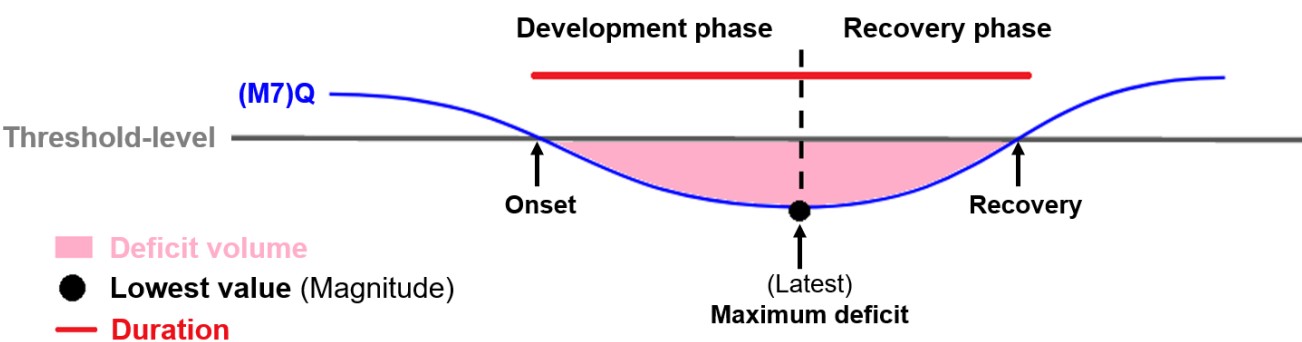

**Figure 6.** Schematic depiction of the event definition and phase subdivision. The extracted (streamflow) drought phases are characterized by duration, event start (onset), the latest date of the maximum streamflow deficit (anomaly), and event recovery. Additional characteristics are the drought intensity (deficit volume or accumulated deficit) and severity/magnitude (maximum streamflow deficit). The computation of other characteristics is left to the user.

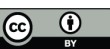



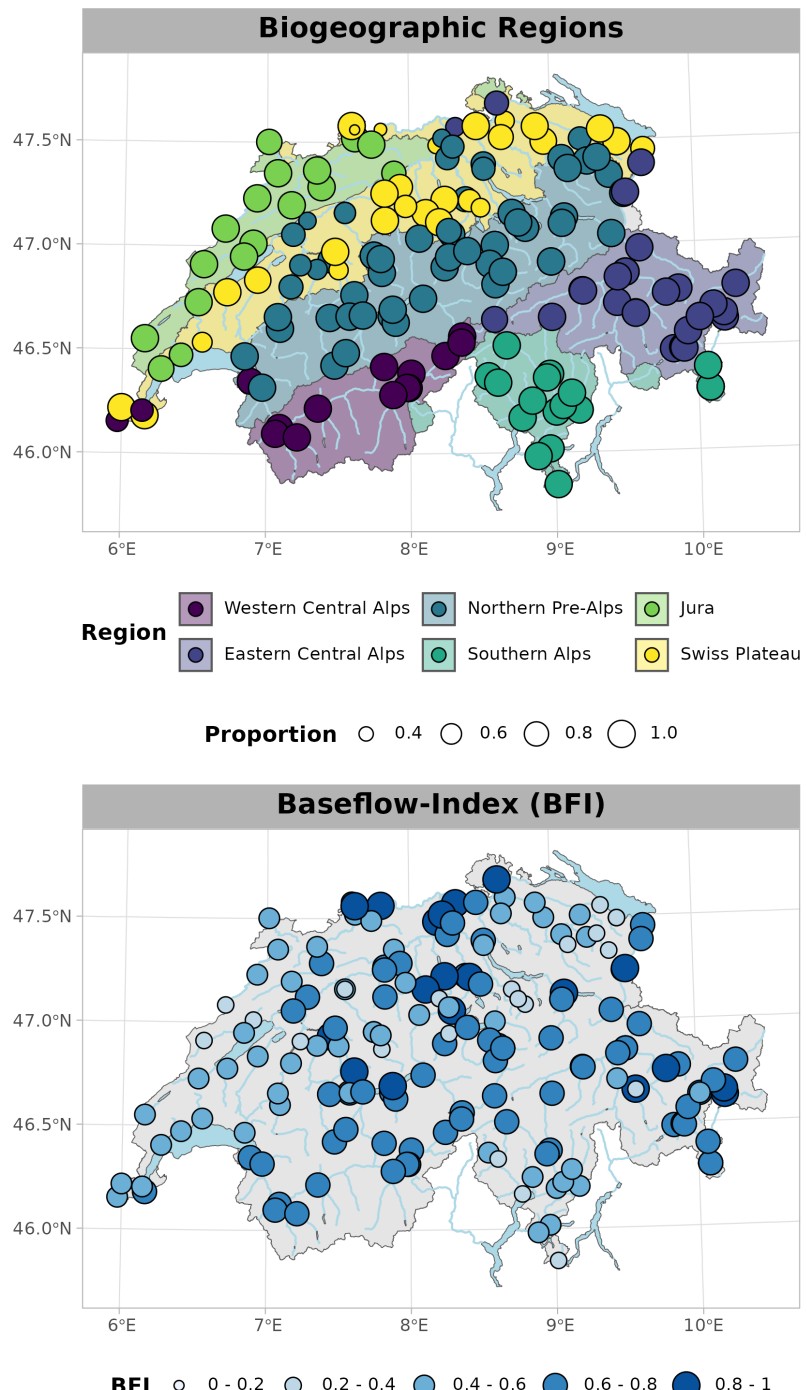

**Figure 7.** Catchment descriptors (examples). **Top**: Dominant (largest overlap percentage with the catchment area) biogeographic region (colours). Point sizes indicate the catchment area proportion covered by the dominant biogeographic region. **Bottom**: Baseflow-Index (BFI, Nathan and McMahon, 1990) for each catchment derived from the daily streamflow time series.



**Figure 8.** Catchment coverage fractions for all (sub-)categories of the soil characteristics: soil depth, skeletal content, water logging, soil permeability, and water storage capacity across regionalized catchment groups derived from the biogeographic regions of Switzerland (colours).

Earth System
Science
Data

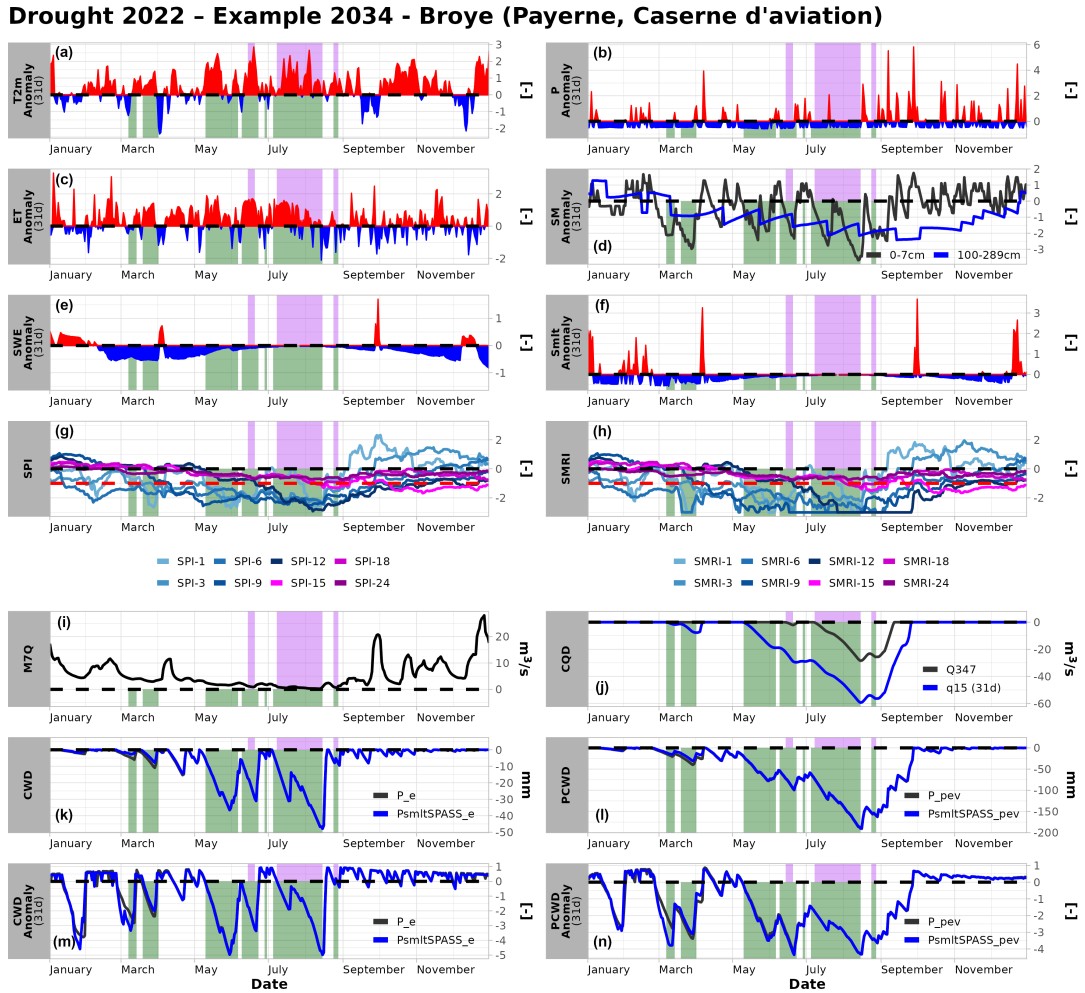

**Figure 9.** Hydro-meteorological time series for the Swiss Plateau catchment 2034 - Broye, Payerne (Caserne d'aviation) for the year 2022. Color shadings in all panels highlight drought periods based on two definitions: yearly Q347 (pink, fixed threshold approach) and a moving monthly $15^{th}$ percentile threshold (green, variable threshold approach). **(a)** Moving monthly anomalies of the 2 m-temperature (T2m), positive anomalies are shown in red and negative anomalies in blue. **(b)** Moving monthly anomalies of the precipitation (P, RhiresD) **(c)** Moving monthly anomalies of the evaporation (ET, ERA5-land). **(d)** Moving monthly anomalies of the soil moisture volume (ESM ERA5-land), soil moisture anomalies are depicted for a near-surface SM-level (black, 0–7 cm) and the deepest level (blue, 100–289 cm) available from ERA5-Land. **(e)** Moving monthly anomalies of the snow water equivalent (SWE SPASS). **(f)** Moving monthly anomalies of the snowmelt (smlt, SPASS). **(g)** SPI colored by aggregation scales from 1- to 24-months. **(h)** SMRI colored by aggregation scales from 1- to 24-months. **(i)** Seven day average streamflow (M7Q). **(j)** The CQD time series shows the corresponding accumulated M7Q-deficits for both the fixed threshold approach (black) and the variable threshold approach (blue). **(k)** Absolute cumulative water deficit (CWD). **(l)** Potential cumulative water deficit (PCWD). **(m)** Monthly anomalies of the CWD (CWD anomaly). **(n)** Monthly anomalies of the PCWD. Time series of the cumulative water deficits for both absolute values and monthly anomalies are shown for both standard (black, P–E (P_e)) and snowmelt-corrected (blue, P–E+ΔSWE (PsmltSPASS_e)) variants. The same is shown for cumulative potential water deficits which are based on the potential water balance (P–PET (P_pev) and P–PET+ΔSWE (PsmltSPASS_pev)).



**Figure 10.** Median SPI values during hydrological drought conditions for all events of all catchments for six selected streamflow regime types across the four seasons winter (DJF), spring (MAM), summer (JJA) and autumn (SON). The streamflow regime types were selected to represent catchments with (dominant) glacial (a-glaciaire, nivo-glaciaire), snow (nival alpin, nival méridional) and pluvial processes (pluvial jurassien, pluvial supérieur) and spatial diversity. Hydrological drought events were defined by a moving monthly (31d) 15th-percentile (variable) threshold. Boxplots are coloured according to SPI aggregation time scales (1- to 24-months). Moderate drought conditions are indicated by the red dashed lines, the black dashed line indicates 0.



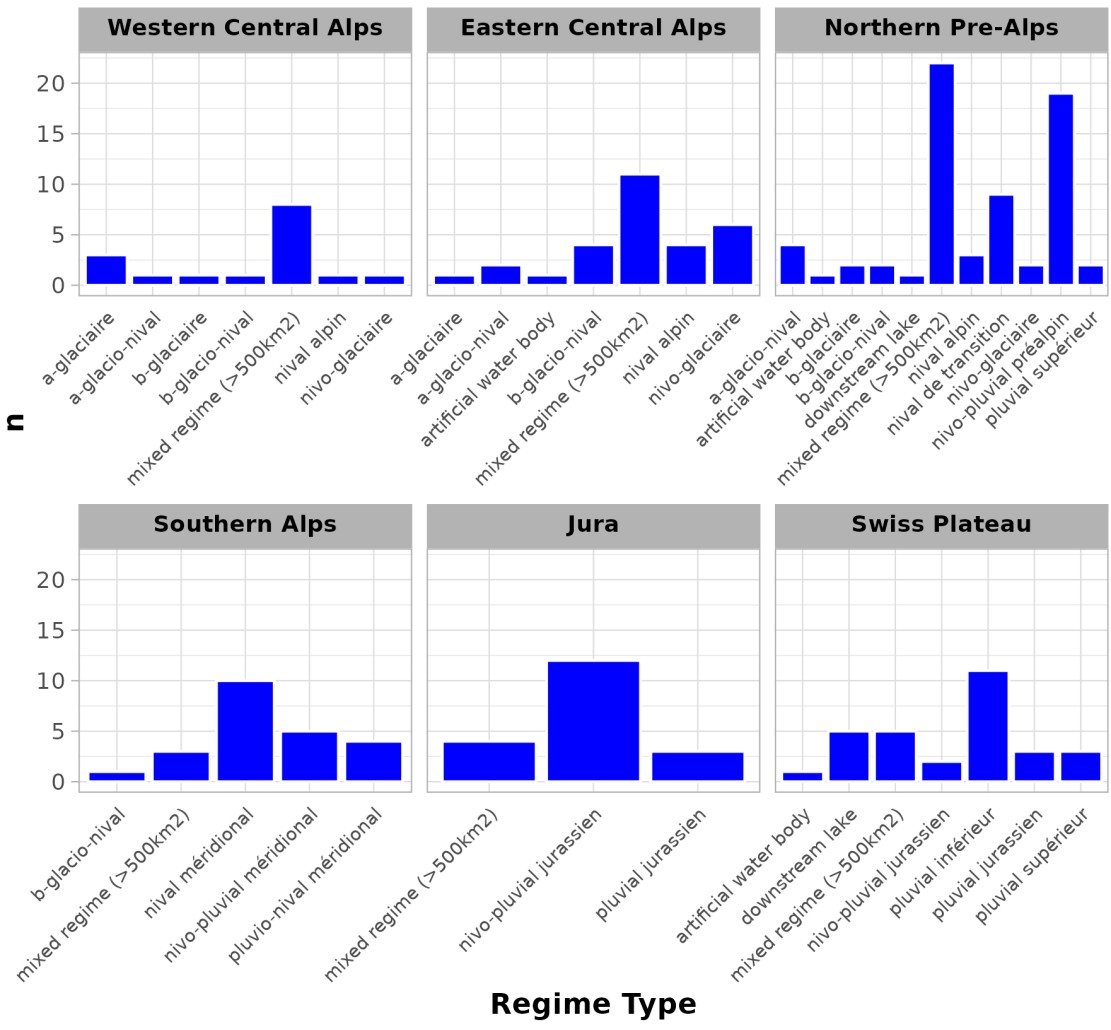

**Figure A1.** Streamflow regime type incidence among catchments grouped by the biogeographic regions of Switzerland (Western Central Alps, Eastern Central Alps, Northern Pre-Alps, Southern Alps, Jura and Swiss Plateau region; see Section 3.3 and also Fig. 7). The streamflow regime type classification was provided by the FOEN.