# Peer review of "HYD-RESPONSES: daily hydro-meteorological catchment-level time series to analyse HYDrological drought dynamics in RESPONSE to (cumulative) water deficits in Swiss catchments."

_Earth System Science Data, 2025_

## Author Comment (AC1)

We would like to thank both reviewers for the careful review and the very useful comments and questions. **Our detailed replies to reviewer 1 start on this page, our detailed replies to reviewer 2 start on page 10.**
Both reviewers ask for additional clarifications and explanations and also for additional user guidance, which we will include. Reviewer 2 points out the paper is quite long and suggests to move the use cases section into the appendix. Since the paper becomes even longer with the additional clarifications, we will follow the suggestions of Reviewer 2 and move the use cases to the appendix. Both reviewers ask for additional guidance on the uncertainty level of the indicators. To address this request and to provide guidance, we will introduce three variables classes (L1 = direct observations and slight processing (e.g spatial averages, L2 = variables from published data sets and slight processing, L3 = derived variables, obtained from input variables from different data sources). more detailed responses are included below and are highlighted in blue, already implemented changes in the manuscript are highlighted in red.

**1 Comments reviewer 1**

This is a review for ESSD, von Matt et al., 2025: HYD-RESPONSES: daily hydro-meteorological catchment-level time series to analyse HYDrological drought dynamics in RESPONSE to (cumulative) water deficits in Swiss catchments.

The described dataset appears to be of potential use, albeit limited for Switzerland-focused usage. I think the manuscript requires major revisions in terms of structure, motivation for certain choices or clearer descriptions as summarized in the general and the specific comments below.

**General/major comments:**

() Item 1. The abstract and the introduction, while motivating and describing the dataset, do not lay out the logic of the manuscript. For example, the use cases in sec 6 come a bit as a surprise to the reader.
We will add a brief outline of the paper structure at the end of the introduction

() In general, the manuscript would benefit from an additional verification effort, showing that the aggregation procedures work as they are intended. For example, 4.2.1: FOEN uses M7(Q), too, so there would be scope for a comparison.
We will add more details on the aggregation procedures in the appendix
() Item 2. Description of catchments and respective coverage by the datasets (section 2-3, figure 1, 2, 3): Here, I think that a bit more care could be given to a comprehensible presentation:

(x) Section 2 would benefit from clarifications i) that the catchments lie partially outside of CH, ii) whether they are fully disjunct (i.e. there is or there is no hierarchy in the catchments), and iii) that any altitude discussed here is basin average.

We added the following clarifications and additional information in Section 2:

**iii):**
In terms of mean catchment height, the catchments are distributed relatively equally between 500

and 2500 m a.s.l. with fewer catchments at elevation ranges above 1500 m a.s.l.

**i) and ii):**

Several catchments contain areas outside of the Swiss national borders as the dataset contains catchments of the entire hydrological Switzerland (catchments that drain in(to) Switzerland). Furthermore, the Swiss streamflow monitoring network is designed such that multiple measurement stations may be located along the same river. As a result, upstream catchments can be nested within larger downstream catchments, leading to hierarchical dependencies.

(x) Then, in section 3.1, there is an explanation why certain stations are not used for the HYD-RESPONSES dataset; some items are self-explanatory, but others are not. Suggest to explain.

We rephrased the reasoning for station inclusion/exclusion and added clarifications on what meaningful means. And yes, the reviewer is absolutely correct that streamflow (Q) is derived from water-level measurements by P-Q relationships. We rephrased the misleading statement for this exclusion criteria. Further, we spelled out what NAWA is (formerly NADUF).

Here, meaningful is equivalent to stations which provide reliable streamflow (Q) time series and are associated with a physical/natural catchment. Stations have therefore be excluded if they i) only provide water-level information (no Q, 3 stations), ii) are not part of the main streamflow measurement network (e.g., stations from other networks such as the National Surface Water Monitoring Programme (NAWA); (BAFU, 2023), 4 stations), iii) secondary stations (11 stations), iv) stations with potential return streamflow (= negative Q values, 2 stations), v) Q measured at derivations (2 stations), vi) stations without watershed delineation (i.e., subterranean; 1 station) and vii) uncertainties in time series composition due to displacement and/or temporarily missing Q of contributing stations (4 stations).

(x) Section 3.2 starts a bit sloppy, as meteorological variables are indeed provided from multiple sources; suggest to re-phrase or re-organise this description. Which data does MeteoSwiss use for RhiresD in areas outside CH?

We rephrased Section 3.2 by adding an introductory paragraph and made the multiple-source coverage of variables clearer. Measurement networks outside Switzerland are not explicitly mentioned in the MeteoSwiss documentation for RhiresD except for that they origin from weather services in neighbouring countries. The added introductory paragraph is as follows:

Hydro-meteorological variables used in this study were compiled from multiple complementary data sources, combining station-based spatial climate analyses, dedicated snow model products, and re-analysis data . This multi-source approach allows both comprehensive coverage of relevant variables and comparative analyses between different data products.

(x) The very last sentence of section 3.2 explains that ERA5-Land data are aggregated to daily values, but does not provide details. Is it done in a way that makes respective data comparable to the other sources (TabsD, TminD, TmaxD, RhiresD)?

We will add clarifications.

() Fig 2 is somewhat unmotivated and only referenced in section 6.2.

We will move the figure to the appendix.

() Section 3.3 does not provide information on coverage outside of CH. This should be clarified. If restricted to CH, is this a problem for catchments with significant portions in neighbouring countries? L344 onwards and the discussion and conclusion sections in principle suggest so.
We will add these clarifications.

(x) Item 3. The information around homogeneity/breakpoints is taken from elsewhere. Still, I think it is important to clearly explain their concepts (L189), because these are not self-explanatory terms. Their usage is also not quite clear. L213: what makes a breakpoint "significant"? Is it an ad hoc definition?

We clarified the concepts and references by adding the following additional information on L189:

Time series homogeneity was derived by the FOEN using breakpoint tests following the method of Bai and Perron (1998). Breakpoints are found by partitioning the time series based on number of potential breakpoints and subsequent modeling of the time series by piecewise linear regression (Bai and Perron, 1998). The optimal breakpoints are found by minimization of the sum of squared residuals. Resulting breakpoints are indicative of changes in the mean annual 7d mean flow (M7Q) and were manually checked by the FOEN based on catchment history and known (potentially) relevant anthropogenic influences such as the construction of (reservoir) dams, hydropower and wastewater treatment plants (for more information see (BAFU, 2024)).

We further added a reference back to adjusted information on L213:

Streamflow time series are provided for three different catchment-specific time periods: 1) the original time series (entire period), 2) the most recent gap-free time-period time series and 3) the most recent homogeneous time series (in case of significant and plausible breakpoints; otherwise equal to the gap-free time series) . The breakpoint information is provided by FOEN .

() Item 4. Guidance on how to use the many different quantities in the dataset; the authors have compiled many different indices, deficit manifestions, etc, and their genesis and differences among related ones are explained well enough. However, what is missing is guidance for users which to use for which purpose. Some examples:

() L245: Can the authors give context on whether the deficit remaining uncompensated over several years is realistic? They provide an extra quantity of annually reset deficits which indicates that it might not be. An expert user will probably know what to make of it, but less experienced users might be a bit lost.
We will add a short discussion.

() Section 4.4: Which application would require the usage of SPI/SPEI/SMRI, respectively?
We will add a short discussion but discussing all possible use cases is beyond the scope of this data paper.

() Section 4.6: fixed and variable threshold definition.
We will add a short comment.

(x) Section 5.2, L335: two BSI variants.

We added additional information for guidance on the use of the two BSI variants:

The first variant is derived based on a ratio between area and length $(A/L^2)$ known as form factor and the second variant is based on a ratio between the catchment area and the area of the circle with the smallest radius encircling the entire catchment $(A_{catch}/A_{circle})$ known as circularity ratio . Both indices range from 0 to 1. Both are frequently used (also in combination) as morphometric catchment indicators . Albeit providing similar information, the form factor is primarily controlled by basin length and hence provides information on catchment elongation while the circularity ratio is more sensitive to basin shape (accounting for complex/irregular shapes resulting in larger area) for more information on basin shape indices see Das et al. (2022).

Item 5. The manuscript remains sometimes short when in comes to motivating certain choices or methods. For example:

(x) L295: Why are more percentiles provided only for M7Q?

We added some accompanying information on the additional percentiles:

For the 7-day average streamflow series (M7Q) we also provide the $2^{nd}$, $10^{th}$ and $15^{th}$ percentiles which are frequently used in streamflow drought analysis.

(x) L309: purpose of "minor pooling"

We added the following information on the purpose of the pooling:

A minor pooling of hydrological drought events is introduced by using the 7-day average streamflow (M7Q) time series, which merges closely successing and potentially dependent individual events to one single event as a result of the smoothing of large day-to-day fluctuations (Tallaksen et al., 1997; Tallaksen and Van Lanen, 2004; Hisdal and Tallaksen, 2000; Sarailidis et al., 2019).

() I suggest to screen the manuscript for these instances and include respective explanations.
We will do so.
() Item 6. Figures and references to figures. I think these must be clearly improved:
We will carefully check that figure references are included where needed.

() Figure 2 was mentioned in my item 2 already.
(x) I find no reference to Figure 6 (although it might help explain the "phases" question that I put under "specific comments"). () Figure 7 should probably specify subfigures as (a) and (b).

() Figure 9: Shouldn't the caption specify *streamflow* drought for the pink and green shadings? I think what is missing here is the time series that was used for defining the drought periods. It seems related to panel j but not identical according to the text from L418 onwards. The y-axes should be labeled properly (T2m: [K], SWE: [mm], ...).
We will revise figure 9.

() Figure 10: not referenced in the main text; the text supposed to discuss the figure mentions "pluvial inferieur", which is not shown in Figure 10, several times around L470. What does the "n" stand for, catchments?
We will revise figure 10.

**Specific comments:**

(x) L23/L89: please introduce CAMELS-CH to the uninformed user not only in section 8 but already at first mention/in the introduction.

We added more contextual information on the CAMELS-CH dataset in both cases.

**L23:**
The dataset is compatible with the recently published "Catchment Attributes and MEteorology for Large-sample Studies" dataset for hydrological Switzerland (CAMELS-CH) and with additional catchment descriptors provided by the FOEN.

**L89:**
The HYD-RESPONSES dataset can be combined with existing hydro-meteorological time series datasets and catchment descriptors such as CAMELS-CH (Höge et al., 2023)

(x) L39/41: do the authors really mean "anthropogenic" in the sense of "caused by humans"? Or is it more about these events impacting the human "sphere" differently? Might also be relevant elsewhere (L375, L390).

We adjusted the formulation in L39/41 and checked other instances to stress that "anthropogenic" refers to anthropogenic (catchment) disturbances which may alter drought (and) streamflow characteristics. Altered characteristics can ultimately also affect the impacts on the human "sphere".

**L39f:**
Individual drought events may differ in their hydro-climatological, hydro-meteorological, hydro-terrestrial and anthropogenic drivers (Mishra and Singh, 2010; Hao and Singh, 2015; Floriancic et al., 2020; Zhou et al., 2021; Massari et al., 2022; Brunner et al., 2023). The consideration of multiple hydro-climatic, hydro-meteorological, hydro-terrestrial and anthropogenic (disturbance) factors is therefore key to understand catchment-specific drought responses and sensitivities and to provide information for drought early warning, preparations, and interventions (e.g., Apurv et al., 2017; Apurv and Cai, 2020; Baez-Villanueva et al., 2024; Brunner et al., 2022, 2021; Ding et al., 2021; Peña-Angulo et al., 2022; Peña-Gallardo et al., 2019; Sutanto and Van Lanen, 2022; Tijdeman et al., 2018; Van Lanen et al., 2013; Savelli et al., 2022; Van Loon and Laaha, 2015; von Matt et al., 2024).

(x) L79: "as a result of non-transformation", is this simply replicating the statement of "non-standardisation"? If so, recommend to delete.

We deleted the replication.

(x) Caption of Table 1: refer to glossary table in the appendix for variable names?

We added the reference to the glossary table

Spatially gridded) products used for the time series extraction. A glossary for all variables can be found in Table XY .

(x) L230-233: The causality here is not entirely clear. I am guessing: The reset at sept 1st is a feature of SPASS SWE in order to avoid the "snow tower" feature. This feature leads to large snowmelt values in delta SWE. These large values are mitigated by the described interpolation. If so (or even if not), the chain could be spelled out more clearly.

Thank you for pointing out that the causality was not sufficiently clear in the original description. We agree and have revised the text to clarify that the large negative ΔSWE values around September

1 are not caused by physical snowmelt processes, but are an artifact of the SPASS model setup. Specifically, SPASS resets modeled SWE at the beginning of each snow year (September 1) to prevent unrealistically large snow accumulation ("snow towers"). When daily changes in SWE ($\Delta$SWE) are computed, this reset produces an artificial negative spike that would otherwise be misinterpreted as extreme snowmelt.

To avoid propagating this known modeling artifact into derived snowfall and snowmelt estimates, $\Delta$ SWE values on September 1 were treated as invalid and replaced by linear interpolation using the adjacent days. This correction affects only a single day per year and serves solely to remove a non-physical signal introduced by the model bookkeeping rather than representing a real hydrological process. The manuscript has been revised accordingly to make this causal chain explicit.

The clarified paragraph is now as follows:

Note that SWE is reset in the SPASS dataset at the end of every snow year (every September $1^{st}$) to avoid unrealistically high accumulation of snow water equivalents ("snow towers") (see Michel et al., 2023). As snowfall and snowmelt were derived from daily differences in SWE ($\Delta$SWE), this reset can result in an artificial large negative $\Delta$SWE value on September $1^{st}$ that does not represent actual physical snowmelt. To prevent this model artifact from affecting the derived snowmelt time series, $\Delta$SWE values on September $1^{st}$ were set to missing values and replaced by linear interpolation using the $\Delta$SWE values from the preceding and following days.

(x) L274: 50 implausible/missing values: Is this referring to the time series, or the distribution (i.e., before or after binning)?

We added some additional information on what missing/implausible values is referring to:

The distribution was selected among the distributions satisfying the following conditions: 1) the transformed values are not significantly different from a normal distribution for the majority of catchments ($p$-values$> 0.05$ for at least 75 % of the catchments), 2) fewer than 5 DOYs flagged and 3) fewer than 50 implausible and/or missing values in the transformed time series (combined consideration of missing values due to flags and unrealistically high/low values).

(x) L282: This means that values are capped at +-3 STD, which could be spelled out here. Can the authors briefly repeat the reasoning for this by Stagge et al. (2015) in this context?

We added the reasoning of Stagge et al. (2015) in this paragraph, specifying implausible values more clearly:

Implausible values are defined as values above or below $+3$ $(-3)$ STD following (Stagge et al., 2015). Estimating more extreme standardized index values from a 30-year climatology requires substantial extrapolation of the fitted distribution and is therefore associated with large uncertainty, particularly given the strong temporal autocorrelation of drought indices. Values beyond 3 correspond to events with return periods far exceeding the length of the reference record and cannot be robustly quantified (Stagge et al., 2015).

(x) L284: Please explain "standard time series".

We rephrased the starting paragraph and specified what is meant by standard time series:

Climatologies and anomalies are provided for all time series, including the standard (raw) time series of extracted variables, derived indicators , standardized drought indices , and cumulative water deficits . The standard time series correspond to the data directly extracted from gridded products or original measurement stations (streamflow), with no modifications except for catchment and temporal aggregation where required .

(x) L299: "Q347" is only introduced in section 5.2; an explanation is necessary at the first instance, I think.

We introduced Q347 in this paragraph but also clarified its derivation in section 5.2:

**L299:**
For the fixed threshold definition, daily M7Q anomalies were derived for events exceeding the Q347 threshold, defined as the daily flow rate exceeded for 347 days per year (i.e., the 347-day exceedance flow, roughly corresponding to the $5^{th}$ streamflow percentile).

**Section 5.2:**
The Q347 (Aschwanden, 1992; Aschwanden and Kan, 1999) is a low flow index used as the basis for water abstraction restrictions in Switzerland and corresponds to the daily flow rate exceeded for 347 days per year. The Q347 was derived from the flow duration curve (FDC) by using the *hydroTSM* R-package (Zambrano-Bigiarini, 2020) and roughly corresponds to the $5^{th}$ streamflow percentile ($95^{th}$ percentile of 365 days $\approx$ 347, hence Q347).

(x) L305: What constitutes a "phase" as used throughout 4.7? Probably related to Figure 6 (unreferenced, see above). Unless this is clarified, it remains unclear what an "event" is.

We added the reference for Figure 6 in section 4.7 and elaborated more on how the events are defined and represented in the event time series:

Streamflow events were also extracted for the fixed (yearly) Q347 threshold An event starts on the first day values fall below the percentile/threshold value and lasts until values exceed the threshold again . For each variant, the event time series consists of consecutively numbered event phases and information on the event duration since the start (i.e, an event of the duration of 5 days may be represented in the time series as: "1 1 1 1 1" (event phase number), "1 2 3 4 5" (duration since start)). Additional event characteristics (e.g., lowest value during a phase) can easily be derived by the user in combination with the indicator time series (an example is provided in the accompanied tutorial on Github; see *Code and data availability*).

(x) Sub-Headings in section 5: "extraction of catchment descriptors (CD)", "derivation of other CD"; these seem not very precise. What exactly is the difference between CD in 5.1 and 5.2? Can the header name or the distinction be specified?

We restructured the catchment descriptor section from two to three subsections:

**1)** Field-based descriptors
**2)** Feature-based descriptors
**3)** Time series-based descriptors

() L331: Is competing areal extent of several categories in a catchment the only possible constraint to representativeness? I could imagine that for example the spatial distribution in terms of

upstream/downstream/margins could also be a factor.
We will add a short discussion

() L365: "partly on both monthly and yearly time scales", this should be specified and motivated.
We will add a short discussion

() L383: The authors have provided a short interpretation of Fig 8; I think they should also do this for Figure A1 (which could also be moved from the appendix to the main part?).
Following the recommendation of Reviewer 2 we will move all case study figures and parts of their discussion to the appendix

() L413: "lacking snowmelt": wouldn't there be more concrete evidence for this somewhere in the HYD-RESPONSES dataset?
We will add a short discussion

(x) L424: "Larger CWDs during..." is this a general (climatological) statement? The term "seasonal climatology" and the "(not shown)" addition might indicate this, but I recommend to stress this more clearly.

We rephrased the sentence to clarify the reference to climatology:

Based on the seasonal climatology of both temperature and evaporation (highest values during summer), larger absolute CWDs are generally expected to occur during the warm season (not shown).

(x) Starting sentence of Section 7: something is not quite right here with the references/model names.

We adjusted the references accordingly and changed the LaTeX-commands from *citet* to *citep*.

**Technical/editorial comments:**
We will implement all editorial suggestions

() L9 onwards: inconsistent in naming all MeteoSwiss/SLF parameters, but only ERA5-Land as a whole.
(x) L12, "information on precipitation, evaporation-driven and streamflow deficits", can the authors please re-phrase; something seems not quite right here.

We rephrased this sentence as follows:

Deficits related to precipitation, evaporation, and streamflow are quantified using both standardized and non-standardized (drought/deficit) indices.

() L93/94: "n=18/94" I think it is misleading as the n refers to streamflow regime types first and then to individual catchments; why not simply include the numbers in prose?
(x) L203, "variables" should go after "flux"?

We adjusted the section to be more precise in terms of variable distinction and conventions used:

For this, instantaneous variables and variables representing accumulations or fluxes are distinguished. For instantaneous variables, we provide daily average values. For variables representing

accumulations and fluxes, we provide daily sums. Flux variables (mainly precipitation and evapo-transpiration) are aggregated using the same temporal convention as RhiresD precipitation sums, i.e., from 06 UTC (day) to 06 UTC (day + 1) (see MeteoSwiss, 2021a). Instantaneous variables and ERA5-Land temperature were averaged from 00 UTC to 00 UTC again following the convention used in equivalent MeteoSwiss products (e.g., TabsD; MeteoSwiss, 2021b).

(x) L264: suggest to unify reference to R packages across the manuscript. E.g., elsewhere it is "cwd R-package", here it is "SCI-package".

We unified all package mentions to *packagename R-package*.

(x) L284: There is a left-over bracket, probably to be closed after "3.2" in L285. done
(x) L293: I suggest to spell out the z score more clearly instead of the "(value-mu)/sigma termi-nology.

We reformulated the sentence as follows:

Using the moving window climatology, standardized anomalies have been derived by first subtract-ing the climatological mean ($\mu$) and then dividing by the climatological standard deviation ($\sigma$) (also known as z-scores: $(value - \mu)/\sigma$).

(x) L308: "event definition", drop "definition"?

We changed the "event definition" to "variant".

For each variant, the event time series consists of consecutively numbered event phases and infor-mation on the event duration since the start.

() L362: MAM = march april may as elsewhere? In general, it is not clear what the acronym(s) is/are supposed to say.
(x) L383 An other > Another changed
(x) L424: "actual" in the sense of "non-anomaly" or in the sense of "non-potential"? Same for L431.

We slightly reformulated both sentences and explanations and also added subfigure references for clarity:

**L424:**
Similar to the longest streamflow drought phases, also the largest cumulative deficits in water balance (CWD) occurred between May–August 2022
**L431f:**
Cumulative deficits in potential water balance (PCWDs ) are more similar to cumulative streamflow deficits for the variable-threshold definition (CQD) . This reflects the different nature of CWDs and PCWDs. Deficits based on the actual water balance (P-E) are more strongly tied to the actual water availability and hence the individual streamflow (drought) phases.

(x) L563 and elsewhere: data set or dataset? changed (x) L563: "alsop" typo changed

**2  Comments reviewer 2**

This manuscript presents a comprehensive hydrometeorological dataset that is developed from a range of other, existing, datasets from various sources. This dataset is developed specifically to support assessment of drought and low flow conditions across catchments in Switzerland, and includes time series of a wide range of essential climate variables as well as derived drought indicators (e.g. SPI, SPEI, etc). As this dataset, or rather what I would consider a data collection can support detailed assessment of drought and the drivers of drought (see also use cases presented in the paper), I would think this in line with the scope of the journal.

Overall, the paper is well structured (with comments, see below) and the datasets that have been developed are outlined clearly, including the original data sources and general comments on data quality.

() One of the main concerns I have is that the dataset combines various underlying sources, in particular observational datasets and re-analysis datasets and datasets from models. Some attention is given at the start of the discussion on where care should be taken, but this is only discussed for selected variables, particularly related to snow (e.g. SWE), but much less discussed for other variables, for example E and PET (see also comments below). In this sense the discussion is somewhat poorly developed. The first part addresses some of the limitations, but a broader reflection on the quality of the dataset, including derived variables, would add to the depth of assessment of what is presented.
This is an important point and we will extend the discussion accordingly

() Perhaps some indicative confidence level on the different datasets and combinations would be useful. I think this should also be added to the metadata provided with the dataset in Zenodo.
Thank you for this suggestion, we will introduce three variable classes (L1 = direct observations and slight processing (e.g., spatial averages, L2 = variables from published data sets and slight processing, L3 = derived variables, obtained from input variables from different data sources)

() The dataset has been developed specifically for Switzerland, and for application in the Swiss context. It would be interesting to comment on how applicable the methods used/presented here would be applicable in other contexts. Some reflection on how applicable this could be in other contexts/countries, what the requirement are of underlying datasets would be etc. For example, use is made of ERA5-Land, which is of course available globally, but on the other hand the availability of observational data in CH is excellent. In other settings where there is less observational data, more use would then perhaps be made of re-analysis data. However, would this still make "sense"?
This is indeed an important point and we will extend the discussion accordingly

() The use cases that are presented in the paper are interesting and indeed demonstrate the utility of the dataset. On the other hand, they may not be the core of the paper, and the paper is very long One could consider including these in the supplementary material. What could then be interesting is to provide some general reflection on the application of these use cases, and how the HYD-RESPONSES dataset and methods have enabled these analyses, and why that was difficult or not possible prior to the development of this dataset.
We will move the use cases and their detailed discussion to the appendix

**Detailed comments:**

(x) Line 98-99: Catchments are described as being at a certain elevation. This seems to be the mean elevation (this is mentioned later). May be useful for comprehension to name that here.

We added the following clarifications and additional information in Section 2:

**iii):**

In terms of mean catchment height, the catchments are distributed relatively equally between 500 and 2500 m a.s.l. with fewer (77 out of 98) catchments at elevation ranges above 1500 m a.s.l. Only eight catchments are higher than 2500 m a.s.l. and only one catchment is at very low elevation (catchment Wiese, Basel).

(x) Line 116: It may be useful to explain a bit better what "meaningful" means. I understand this is obtained through personal communication, but it is somewhat vague.

(x) Line 118: Stations where Q is measure at a water level station. I find this somewhat confusing, as I would presume (and my experience working with FOEN would confirm) that most river stations measure water levels and derive the discharge through a rating curve. Perhaps something else is meant. Please clarify.

(x) Line 118: Mention is made of NADUF stations. I am not familiar with what these are. Please provide some explanation.

We rephrased the reasoning for station inclusion/exclusion and added clarifications on what meaningful means. And yes, the reviewer is absolutely correct that streamflow (Q) is derived from water-level measurements by P-Q relationships. We rephrased the misleading statement for this exclusion criteria. Further, we spelled out what NAWA is (formerly NADUF).

Here, meaningful is equivalent to stations which provide reliable streamflow (Q) time series and are associated with a physical/natural catchment. Stations have therefore be excluded if they i) only provide water-level information (no Q, 3 stations), ii) are not part of the main streamflow measurement network (e.g., stations from other networks such as the National Surface Water Monitoring Programme (NAWA; BAFU, 2023, 4 stations), iii) secondary stations (11 stations), iv) stations with potential return streamflow (= negative Q values, 2 stations), v) Q measured at derivations (2 stations), vi) stations without watershed delineation (i.e., subterranean; 1 station) and vii) uncertainties in time series composition due to displacement and/or temporarily missing Q of contributing stations (4 stations).

() Line 121: list of stations included
We will clarify this

(x) Line 124: May be useful to mention what is meant by assembled – I assume that this is compiling the catchment average for these variables.
We changed the wording from "assembled" to "extracted". But as the detailed methodology is presented in Section 4 (Data processing) we did not replicate further methodological details in this section.

() Line 126: I appreciate that the authors use the original names/ids of data depending on the

source (e.g. RhiresD, tp). Table 1A provides some explanation which is useful. Perhaps it would be useful to provide in that table something like the WMO standard naming conventions. Would also be clear if the authors mention this strategy in the text, as it may otherwise become somewhat confusing.

We will clarify this point

(x) Line 139: Mentions is made of the quantile mapping approach being used. It is not so clear to me what the reference is for this bias correction through quantile mapping. Please clarify.

We slightly restructured the part on SPASS and removed the direct mentioning of quantile mapping (which is part of SnowQM):

The preliminary version was produced in 2022 and provides modeled and bias-corrected daily SWE data for the period September 1961–September 2022. The underlying SnowQM model is presented in detail in Michel et al. (2023). SWE is derived based on the daily TabsD and RhiresD products (see above). The spatial extent is restricted to the Swiss territory.

(x) Line 167: I think the word "The" at the start of the sentence needs to be dropped as it is otherwise not clear which digital soil suitability maps are intended as these have not been introduced.

We added "of Switzerland" to the sentence to make clear that we refer specifically to the maps used in the HYD-RESPONSES dataset which are introduced with this sentence.

The digital soil suitability maps of Switzerland provide information on a set of different soil characteristics assessed on 25 different geological and geomorphological units which are further discriminated by different landscape elements depending on aspect, slope and bedrock.

() Line 192: Does mean height here imply the mean elevation? Please be consistent.
We will clarify

(x) Line 205: Check the sentence starting with "For accumulation... ". It is somewhat confusing and may need to be rephrased or elaborated to be clear.

We rephrased the sentence as follows:

For variables representing accumulations and fluxes, we provide daily sums.

(x) Line 215-216: Limnographs are often mentioned. The word is correct but to my mind not in such common use. It is somewhat a Germanism to my mind. Perhaps use Water level sensor or something similar
We have changed the name to your suggestion from limnograph to water level sensor.

(x) Line 233: Mention is made of interpolation between the day before the end of September and the day after, when SWE is det to zero. Surely the amount of water that melts is the same whether this happens over one or two days – and still unrealistically high, given that when resetting I assume all snow is considered as being melted. Perhaps I am misunderstanding the concept.

We clarified the paragraph as follows:

Note that SWE is reset in the SPASS dataset at the end of every snow year (every September $1^{st}$)

to avoid unrealistically high accumulation of snow water equivalents ("snow towers") (see Michel et al., 2023). As snowfall and snowmelt were derived from daily differences in SWE ($\Delta$SWE), this reset can result in an artificial large negative $\Delta$SWE value on September $1^{st}$ that does not represent actual physical snowmelt. To prevent this model artifact from affecting the derived snowmelt time series, $\Delta$SWE values on September $1^{st}$ were set to missing values and replaced by linear interpolation using the $\Delta$SWE values from the preceding and following days.

() Line 239: Also related to the general comment. Here E and PET are derived from observed P (interpolated) and E and PET calculated in ERA5-Land. I am not clear how biases are dealt with, especially in E. If in ERA5-Land the precipitation is strongly biased, then surely E will (climatologically) tend to be too low in catchments that are water limited.
We will add a short discussion

(x) Line 246: Here it seems to be suggested that PCWD is calculated using ERA5-Variables. Is that both for P and PET? Please clarify.

Yes, the HYD-RESPONSES dataset also provides PCWDs based on ERA5-LAND variables only (i.e., tp−pev). We have specified this in L246 as follows:

In some cases (especially for P–PET based only on ERA5-Land variables, i.e., tp−pev), PCWDs are not compensated each year and can persist over multiple years.

(x) Line 255: The word period is used here to indicate the accumulation window for SPI and SPEI. In other sections the word period is used to denote a period in time (e.g. 10 years). Please use words consistently with a defined meaning, as it is otherwise somewhat confusing.

We changed the wording for standardized indices to aggregation (time) windows and to events where applicable.

(x) Line 273: mention is made of fewer than five DOYs flagged. I am not sure how these are flagged! Perhaps I missed it.

We clarified how the flags are obtained and what they signify by adding some additional information in the section:

The suitability of candidate distributions was assessed based on three indicators: the Shapiro-Wilks normality tests ($p$-values; Shapiro and Wilk, 1965), the number of flags returned by the fitting function *fitSCI* (see SCI R-package; Gudmundsson and Stagge, 2016), and the number of missing and/or implausible values. Implausible values are defined as values above or below $+3$ ($-3$) STD following (Stagge et al., 2015). The returned flags in distribution parameter fitting were mainly related to convergence issues (non-convergence) (flag 3, see SCI R-package Gudmundsson and Stagge, 2016). Without a valid fit, the transformation to standardized index values is not possible resulting in missing values on the flagged DOYs in all time series years.

(x) Line 282: I was curious in the derivation of the distributional parameters of the distributions applied in SPI, SPEI and SMRI, if the same period of data was used to derive the parameters of the distribution, or if for each case the whole available time series was used. That could make comparison more difficult.

Yes, the same period of data was used to derive the parameters of the various distributions. We

added additional clarifications to the section:

The distributions were fitted for each day of the year (DOY) based on the reference period 1991–2020. This results in a fit for each DOY derived from the same (window of) values for each distribution. Monthly SPI fits (SPI-1) are for example based on the 30 daily values up to the specific DOY for each of the 30 years in the reference period 1991–2020.

(x) Line 292: Mai à May
Adjusted.

(x) Line 336: Length – I guess of the main drainage path -please clarify.

we specified what length means as follows:

The first variant is derived based on a ratio between area and basin length $(A/L^2)$ known as form factor and the second variant is based on a ratio between the catchment area and the area of the circle with the smallest radius encircling the entire catchment $(A_{catch}/A_{circle})$ known as circularity ratio.

() Line 421: What does HRSg mean – often used but not clarified.
We will clarify

(x) Line 441: Why the 15th percentile – if this is just to illustrate the please state is an arbitrary threshold.

We added the motivation for the choice of that percentile in the sentence:

Here, we present a composite analysis of median SPI values associated with streamflow droughts defined by the monthly $15^{th}$-percentiles of the streamflow which corresponds to the highest of the low-flow percentile used for the Swiss national drought platform .

() Line 490: It may be good to note that ERA5-Reanalysis and ERA5-Land are (to the best of my knowledge) not independent, with ERA5-Land derived from the former by downscaling using features such as elevation etc.
We will add a brief discussion

() Line 499: The discussion that ends here is relevant, as in the dataset several indicators are developed that combine data from different sources - such as the cumulative water deficit, and the snowmelt corrected precipitation datasets. Given that these combine observational and reanalysis data, this may result in different levels of reliability of the derived datasets. I would be curious as to how is this flagged in these derived datasets. In other words, is some flag of degree of confidence set in the meta-data?
To be discussed

(x) Table 3: I am not sure if Zenodo can be considered a provider – is this not more a repository?
We will change the wording where applicable.

(x) Line 563: also

Changed.

(x) Figure 4: "complete" is mentioned – I guess this is the same as full. Nice to be consistent.
Changed.

() Figure 5: The label NAs/Implausible should be described as to what it means (one can guess of course – but best to be clear).
We will clarify

**References**

[revised manuscript text omitted]

---

## Author Comment (AC3)

We would like to thank both reviewers for the careful review and the very useful comments and questions. **Our detailed replies to reviewer 1 start on this page, our detailed replies to reviewer 2 start on page 10.**
Both reviewers ask for additional clarifications and explanations and also for additional user guidance, which we will include. Reviewer 2 points out the paper is quite long and suggests to move the use cases section into the appendix. Since the paper becomes even longer with the additional clarifications, we will follow the suggestions of Reviewer 2 and move the use cases to the appendix. Both reviewers ask for additional guidance on the uncertainty level of the indicators. To address this request and to provide guidance, we will introduce three variables classes (L1 = direct observations and slight processing (e.g spatial averages, L2 = variables from published data sets and slight processing, L3 = derived variables, obtained from input variables from different data sources). more detailed responses are included below and are highlighted in blue, already implemented changes in the manuscript are highlighted in red.

**1 Comments reviewer 1**

This is a review for ESSD, von Matt et al., 2025: HYD-RESPONSES: daily hydro-meteorological catchment-level time series to analyse HYDrological drought dynamics in RESPONSE to (cumulative) water deficits in Swiss catchments.

The described dataset appears to be of potential use, albeit limited for Switzerland-focused usage. I think the manuscript requires major revisions in terms of structure, motivation for certain choices or clearer descriptions as summarized in the general and the specific comments below.

**General/major comments:**

(x) Item 1. The abstract and the introduction, while motivating and describing the dataset, do not lay out the logic of the manuscript. For example, the use cases in sec 6 come a bit as a surprise to the reader.

We added the outline of the paper structure at the end of the introduction:

The remaining paper is structured as follows: In section 2 the study region and included catchments are presented and described by their basic characteristics. Section 3 introduces all datasets used to compile the HYD-RESPONSES dataset including spatio-temporal raster data of hydro-meteorological variables and datasets used to derive important catchment descriptors. Section 4 elaborates on the processing of hydro-meteorological data consisting of catchment-level time series extraction and derived (drought) indicators, deficit calculation and drought event definition. Section 5 is the analogue for the processing and extraction of catchment descriptors. Section 6 finally discusses the dataset and points to potential caveats and cautionary notes while section 7 presents multiple complementary datasets which are valuable in combination with the HYD-RESPONSES dataset. Section 8 provides a concluding summary on the presented dataset. For the interested reader, three case studies on the potential of the HYD-RESPONSES dataset are provided in Appendix B.

() In general, the manuscript would benefit from an additional verification effort, showing that the

aggregation procedures work as they are intended. For example, 4.2.1: FOEN uses M7(Q), too, so there would be scope for a comparison.

Thank you for the suggestion. For the M7Q, no aggregation was needed except for a smoothing of the original time series by a 7-day moving average window (using the *zoo* R-package). A scope for comparison might be the Q347 low-flow indicator - but then again - the procedure is following the one of the FOEN based on the same (original) FOEN streamflow time series. Here, we however wanted to ensure catchment intercomparabilty by using a common reference period (1990–2010) among all catchments whereas to my best knowledge - the FOEN uses the entire (homogeneous) time series. Hence, slight differences would be expected.

() Item 2. Description of catchments and respective coverage by the datasets (section 2-3, figure 1, 2, 3): Here, I think that a bit more care could be given to a comprehensible presentation:

(x) Section 2 would benefit from clarifications i) that the catchments lie partially outside of CH, ii) whether they are fully disjunct (i.e. there is or there is no hierarchy in the catchments), and iii) that any altitude discussed here is basin average.

We added the following clarifications and additional information in Section 2:

**iii):**
In terms of mean catchment height, the catchments are distributed relatively equally between 500 and 2500 m a.s.l. with fewer catchments at elevation ranges above 1500 m a.s.l.
**i) and ii):**
Several catchments contain areas outside of the Swiss national borders as the dataset contains catchments of the entire hydrological Switzerland (catchments that drain in(to) Switzerland). Furthermore, the Swiss streamflow monitoring network is designed such that multiple measurement stations may be located along the same river. As a result, upstream catchments can be nested within larger downstream catchments, leading to hierarchical dependencies.

(x) Then, in section 3.1, there is an explanation why certain stations are not used for the HYDRESPONSES dataset; some items are self-explanatory, but others are not. Suggest to explain.

We rephrased the reasoning for station inclusion/exclusion and added clarifications on what meaningful means. And yes, the reviewer is absolutely correct that streamflow (Q) is derived from water-level measurements by P-Q relationships. We rephrased the misleading statement for this exclusion criteria. Further, we spelled out what NAWA is (formerly NADUF).

Here, meaningful is equivalent to stations which provide reliable streamflow (Q) time series and are associated with a physical/natural catchment. Stations have therefore be excluded if they i) only provide water-level information (no Q, 3 stations), ii) are not part of the main streamflow measurement network (e.g., stations from other networks such as the National Surface Water Monitoring Programme (NAWA); (BAFU, 2023), 4 stations), iii) secondary stations (11 stations), iv) stations with potential return streamflow (= negative Q values, 2 stations), v) Q measured at derivations (2 stations), vi) stations without watershed delineation (i.e., subterranean; 1 station) and vii) uncertainties in time series composition due to displacement and/or temporarily missing Q of contributing stations (4 stations).

(x) Section 3.2 starts a bit sloppy, as meteorological variables are indeed provided from multiple sources; suggest to re-phrase or re-organise this description. Which data does MeteoSwiss use for RhiresD in areas outside CH?

We rephrased Section 3.2 by adding an introductory paragraph and made the multiple-source coverage of variables clearer. Measurement networks outside Switzerland are not explicitly mentioned in the MeteoSwiss documentation for RhiresD except for that they origin from weather services in neighbouring countries. The added introductory paragraph is as follows:

Hydro-meteorological variables used in this study were compiled from multiple complementary data sources, combining station-based spatial climate analyses, dedicated snow model products, and reanalysis data . This multi-source approach allows both comprehensive coverage of relevant variables and comparative analyses between different data products.

(x) The very last sentence of section 3.2 explains that ERA5-Land data are aggregated to daily values, but does not provide details. Is it done in a way that makes respective data comparable to the other sources (TabsD, TminD, TmaxD, RhiresD)?

We will add clarifications.

() Fig 2 is somewhat unmotivated and only referenced in section 6.2.
We will move the figure to the appendix.

(x) Section 3.3 does not provide information on coverage outside of CH. This should be clarified. If restricted to CH, is this a problem for catchments with significant portions in neighbouring countries? L344 onwards and the discussion and conclusion sections in principle suggest so.

Thank you for pointing this out. We changed the section as follows:

First, we excluded the swissALTI3D from the list of used datasets, as information on aspect and slope can be retrieved directly via complementary datasets (e.g., the accompanying catchment information provided by the FOEN) which are derived from data that cover all catchments entirely. This was not the case for swissALTI3D.

We further completed information on catchment coverage in the last paragraph of the section:
Note that the digital soil suitability maps, swissTLM3D hydrography, biogeographic regions of Switzerland as well as information on springs and swallow holes in karst regions are restricted to Swiss national territory. Catchments with significant area outside of Switzerland may be treated with caution for descriptive variables extracted from these datasets (see section 5 for a comprehensive overview on extracted descriptors). The hydrogeological and lithological maps of Switzerland to a large extent also cover areas outside of Switzerland. Only catchments of the Rhine (*2091, 2143, 2288, 2289*) and Wiese (*2199*) are not entirely covered. However, with a coverage of >94%, descriptors extracted from these datasets may still prove valuable.
Methodological details on the extraction and preparation of catchment descriptors are presented in Section 5.

(x) Item 3. The information around homogeneity/breakpoints is taken from elsewhere. Still, I think it is important to clearly explain their concepts (L189), because these are not self-explanatory terms. Their usage is also not quite clear. L213: what makes a breakpoint "significant"? Is it an ad hoc definition?

We clarified the concepts and references by adding the following additional information on L189:

Time series homogeneity was derived by the FOEN using breakpoint tests following the method of Bai and Perron (1998). Breakpoints are found by partitioning the time series based on number of potential breakpoints and subsequent modeling of the time series by piecewise linear regression (Bai and Perron, 1998). The optimal breakpoints are found by minimization of the sum of squared residuals. Resulting breakpoints are indicative of changes in the mean annual 7d mean flow (M7Q) and were manually checked by the FOEN based on catchment history and known (potentially) relevant anthropogenic influences such as the construction of (reservoir) dams, hydropower and wastewater treatment plants (for more information see (BAFU, 2024)).

We further added a reference back to adjusted information on L213:

Streamflow time series are provided for three different catchment-specific time periods: 1) the original time series (entire period), 2) the most recent gap-free time-period time series and 3) the most recent homogeneous time series (in case of significant and plausible breakpoints; otherwise equal to the gap-free time series) . The breakpoint information is provided by FOEN .

Item 4. Guidance on how to use the many different quantities in the dataset; the authors have compiled many different indices, deficit manifestions, etc, and their genesis and differences among related ones are explained well enough. However, what is missing is guidance for users which to use for which purpose. Some examples:

() L245: Can the authors give context on whether the deficit remaining uncompensated over several years is realistic? They provide an extra quantity of annually reset deficits which indicates that it might not be. An expert user will probably know what to make of it, but less experienced users might be a bit lost.

We will add a short discussion.

() Section 4.4: Which application would require the usage of SPI/SPEI/SMRI, respectively?
We will add a short discussion but discussing all possible use cases is beyond the scope of this data paper.

(x) Section 4.6: fixed and variable threshold definition.

We completed the section with information on the strengths of both fixed and variable percentiles to provide guidance for the user on when to use which type of deficits:

Both fixed and variable thresholds are useful in streamflow drought analysis because they capture different aspects of hydrological droughts. Fixed thresholds (e.g., a constant percentile threshold) are often used to assess absolute water scarcity and are well suited for evaluating impacts on water supply systems, ecosystems, and regulatory limits, where critical flow levels do not change seasonally . In contrast, variable thresholds (e.g., seasonally varying percentiles) account for natural intra-annual variability in streamflow and are better suited for detecting anomalous conditions relative to the expected seasonal regime, thereby targeting hydrological droughts as deviations from normal conditions . Using both approaches allows for a more comprehensive characterization of droughts, linking absolute severity with relative anomalies.

(x) Section 5.2, L335: two BSI variants.

We added additional information for guidance on the use of the two BSI variants:

The first variant is derived based on a ratio between area and length ($A/L^2$) known as form factor and the second variant is based on a ratio between the catchment area and the area of the circle with the smallest radius encircling the entire catchment ($A_{catch}/A_{circle}$) known as circularity ratio. Both indices range from 0 to 1. Both are frequently used (also in combination) as morphometric catchment indicators. Albeit providing similar information, the form factor is primarily controlled by basin length and hence provides information on catchment elongation while the circularity ratio is more sensitive to basin shape (accounting for complex/irregular shapes resulting in larger area) for more information on basin shape indices see Das et al. (2022).

Item 5. The manuscript remains sometimes short when in comes to motivating certain choices or methods. For example:

(x) L295: Why are more percentiles provided only for M7Q?

We added some accompanying information on the additional percentiles:

For the 7-day average streamflow series (M7Q) we also provide the $2^{nd}$, $10^{th}$ and $15^{th}$ percentiles which are frequently used in streamflow drought analysis.

(x) L309: purpose of "minor pooling"

We added the following information on the purpose of the pooling:

A minor pooling of hydrological drought events is introduced by using the 7-day average streamflow (M7Q) time series, which merges closely successing and potentially dependent individual events to one single event as a result of the smoothing of large day-to-day fluctuations (Tallaksen et al., 1997; Tallaksen and Van Lanen, 2004; Hisdal and Tallaksen, 2000; Sarailidis et al., 2019).

() I suggest to screen the manuscript for these instances and include respective explanations.
We will do so.
() Item 6. Figures and references to figures. I think these must be clearly improved:
We will carefully check that figure references are included where needed.

() Figure 2 was mentioned in my item 2 already.
(x) I find no reference to Figure 6 (although it might help explain the "phases" question that I put under "specific comments"). () Figure 7 should probably specify subfigures as (a) and (b).

() Figure 9: Shouldn't the caption specify *streamflow* drought for the pink and green shadings? I think what is missing here is the time series that was used for defining the drought periods. It seems related to panel j but not identical according to the text from L418 onwards. The y-axes should be labeled properly (T2m: [K], SWE: [mm], ...).
We will revise figure 9.

() Figure 10: not referenced in the main text; the text supposed to discuss the figure mentions "pluvial inferieur", which is not shown in Figure 10, several times around L470. What does the "n" stand for, catchments?

We will revise figure 10.

**Specific comments:**
(x) L23/L89: please introduce CAMELS-CH to the uninformed user not only in section 8 but already at first mention/in the introduction.

We added more contextual information on the CAMELS-CH dataset in both cases.

**L23:**
The dataset is compatible with the recently published "Catchment Attributes and MEteorology for Large-sample Studies" dataset for hydrological Switzerland (CAMELS-CH) and with additional catchment descriptors provided by the FOEN.

**L89:**
The HYD-RESPONSES dataset can be combined with existing hydro-meteorological time series datasets and catchment descriptors such as CAMELS-CH (Höge et al., 2023) which provides large-sample hydro-meteorological data for hydrologic Switzerland and is the Swiss version of the "Catchment Attributes and MEteorology for Large-sample Studies" CAMELS; (see e.g., Clerc-Schwarzenbach et al., 2024).

(x) L39/41: do the authors really mean "anthropogenic" in the sense of "caused by humans"? Or is it more about these events impacting the human "sphere" differently? Might also be relevant elsewhere (L375, L390).

We adjusted the formulation in L39/41 and checked other instances to stress that "anthropogenic" refers to anthropogenic (catchment) disturbances which may alter drought (and) streamflow characteristics. Altered characteristics can ultimately also affect the impacts on the human "sphere".

**L39f:**
Individual drought events may differ in their hydro-climatological, hydro-meteorological, hydro-terrestrial and anthropogenic drivers (Mishra and Singh, 2010; Hao and Singh, 2015; Floriancic et al., 2020; Zhou et al., 2021; Massari et al., 2022; Brunner et al., 2023). The consideration of multiple hydro-climatic, hydro-meteorological, hydro-terrestrial and anthropogenic (disturbance) factors is therefore key to understand catchment-specific drought responses and sensitivities and to provide information for drought early warning, preparations, and interventions (e.g., Apurv et al., 2017; Apurv and Cai, 2020; Baez-Villanueva et al., 2024; Brunner et al., 2022, 2021; Ding et al., 2021; Peña-Angulo et al., 2022; Peña-Gallardo et al., 2019; Sutanto and Van Lanen, 2022; Tijdeman et al., 2018; Van Lanen et al., 2013; Savelli et al., 2022; Van Loon and Laaha, 2015; von Matt et al., 2024).

(x) L79: "as a result of non-transformation", is this simply replicating the statement of "non-standardisation"? If so, recommend to delete.

We deleted the replication.

(x) Caption of Table 1: refer to glossary table in the appendix for variable names?

We added the reference to the glossary table

Spatially gridded) products used for the time series extraction. A glossary for all variables can be found in Table A1 .

(x) L230-233: The causality here is not entirely clear. I am guessing: The reset at sept 1st is a feature of SPASS SWE in order to avoid the "snow tower" feature. This feature leads to large snowmelt values in delta SWE. These large values are mitigated by the described interpolation. If so (or even if not), the chain could be spelled out more clearly.

Thank you for pointing out that the causality was not sufficiently clear in the original description. We agree and have revised the text to clarify that the large negative $\Delta$SWE values around September 1 are not caused by physical snowmelt processes, but are an artifact of the SPASS model setup. Specifically, SPASS resets modeled SWE at the beginning of each snow year (September 1) to prevent unrealistically large snow accumulation ("snow towers"). When daily changes in SWE ($\Delta$SWE) are computed, this reset produces an artificial negative spike that would otherwise be misinterpreted as extreme snowmelt.

To avoid propagating this known modeling artifact into derived snowfall and snowmelt estimates, $\Delta$ SWE values on September 1 were treated as invalid and replaced by linear interpolation using the adjacent days. This correction affects only a single day per year and serves solely to remove a non-physical signal introduced by the model bookkeeping rather than representing a real hydrological process. The manuscript has been revised accordingly to make this causal chain explicit.

The clarified paragraph is now as follows:

Note that SWE is reset in the SPASS dataset at the end of every snow year (every September $1^{st}$) to avoid unrealistically high accumulation of snow water equivalents ("snow towers") (see Michel et al., 2023). As snowfall and snowmelt were derived from daily differences in SWE ($\Delta$SWE), this reset can result in an artificial large negative $\Delta$SWE value on September $1^{st}$ that does not represent actual physical snowmelt. To prevent this model artifact from affecting the derived snowmelt time series, $\Delta$SWE values on September $1^{st}$ were set to missing values and replaced by linear interpolation using the $\Delta$SWE values from the preceding and following days.

(x) L274: 50 implausible/missing values: Is this referring to the time series, or the distribution (i.e., before or after binning)?

We added some additional information on what missing/implausible values is referring to:

The distribution was selected among the distributions satisfying the following conditions: 1) the transformed values are not significantly different from a normal distribution for the majority of catchments ($p$-values$> 0.05$ for at least 75 % of the catchments), 2) fewer than 5 DOYs flagged and 3) fewer than 50 implausible and/or missing values in the transformed time series (combined consideration of missing values due to flags and unrealistically high/low values).

(x) L282: This means that values are capped at +-3 STD, which could be spelled out here. Can the authors briefly repeat the reasoning for this by Stagge et al. (2015) in this context?

We added the reasoning of Stagge et al. (2015) in this paragraph, specifying implausible values more clearly:

Implausible values are defined as values above or below $+3$ $(-3)$ STD following (Stagge et al., 2015). Estimating more extreme standardized index values from a 30-year climatology requires substantial extrapolation of the fitted distribution and is therefore associated with large uncertainty, particularly given the strong temporal autocorrelation of drought indices. Values beyond 3 correspond to

events with return periods far exceeding the length of the reference record and cannot be robustly quantified (Stagge et al., 2015).

(x) L284: Please explain "standard time series".

We rephrased the starting paragraph and specified what is meant by standard time series:

Climatologies and anomalies are provided for all time series, including the standard (raw) time series of extracted variables, derived indicators , standardized drought indices , and cumulative water deficits . The standard time series correspond to the data directly extracted from gridded products or original measurement stations (streamflow), with no modifications except for catchment and temporal aggregation where required .

(x) L299: "Q347" is only introduced in section 5.2; an explanation is necessary at the first instance, I think.

We introduced Q347 in this paragraph but also clarified its derivation in section 5.2:

**L299:**
For the fixed threshold definition, daily M7Q anomalies were derived for events exceeding the Q347 threshold, defined as the daily flow rate exceeded for 347 days per year (i.e., the 347-day exceedance flow, roughly corresponding to the $5^{th}$ streamflow percentile).

**Section 5.2:**
The Q347 (Aschwanden, 1992; Aschwanden and Kan, 1999) is a low flow index used as the basis for water abstraction restrictions in Switzerland and corresponds to the daily flow rate exceeded for 347 days per year. The Q347 was derived from the flow duration curve (FDC) by using the *hydroTSM* R-package (Zambrano-Bigiarini, 2020) and roughly corresponds to the $5^{th}$ streamflow percentile ($95^{th}$ percentile of 365 days $\approx$ 347, hence Q347).

(x) L305: What constitutes a "phase" as used throughout 4.7? Probably related to Figure 6 (unreferenced, see above). Unless this is clarified, it remains unclear what an "event" is.

We added the reference for Figure 6 in section 4.7 (note that these references are not included in the text below) and elaborated more on how the events are defined and represented in the event time series:

Streamflow events were also extracted for the fixed (yearly) Q347 threshold An event starts on the first day values fall below the percentile/threshold value and lasts until values exceed the threshold again . For each variant, the event time series consists of consecutively numbered event phases and information on the event duration since the start (i.e, an event of the duration of 5 days may be represented in the time series as: "1 1 1 1 1" (event phase number), "1 2 3 4 5" (duration since start)). Additional event characteristics (e.g., lowest value during a phase) can easily be derived by the user in combination with the indicator time series (an example is provided in the accompanied tutorial on Github; see *Code and data availability*).

(x) Sub-Headings in section 5: "extraction of catchment descriptors (CD)", "derivation of other CD"; these seem not very precise. What exactly is the difference between CD in 5.1 and 5.2? Can the header name or the distinction be specified?

We restructured the catchment descriptor section from two to three subsections:

**1)** Field-based descriptors
**2)** Feature-based descriptors
**3)** Time series-based and climatological descriptors

() L331: Is competing areal extent of several categories in a catchment the only possible constraint to representativeness? I could imagine that for example the spatial distribution in terms of upstream/downstream/margins could also be a factor.
We will add a short discussion.

(x) L365: "partly on both monthly and yearly time scales", this should be specified and motivated.

Thank you for checking this statement. We initially wanted to provide the indices on monthly basis too as to identify seasonal variability in hydro-climatic catchment controls. In the dataset on Zenodo used for the preprint we did however only include yearly information (except for Pardé coefficients). The sentence was adjusted accordingly and reasoning has been added. The provided time series allow the user to derive additional indices on seasonal scale if required.

Information on average precipitation, temperature, evaporation, snow water equivalent, streamflow, the fraction of precipitation falling as snow and the runoff fraction (Q/P) are provided on yearly scales for identifying broad climatic (i.e., water balance) and physiographic controls on hydrological behavior.

() L383: The authors have provided a short interpretation of Fig 8; I think they should also do this for Figure A1 (which could also be moved from the appendix to the main part?).
Following the recommendation of Reviewer 2 we will move all case study figures - except the Fig.9 which is kept for time series illustration but not the in-depth case study analysis - and parts of their discussion to the appendix.

(x) L413: "lacking snowmelt": wouldn't there be more concrete evidence for this somewhere in the HYD-RESPONSES dataset?

Thank you for pointing this out. We added a reference to panels e) and f) in Figure 9 which depict the anomalies in monthly SWE and (SWE-derived) snowmelt anomalies.

During the variable threshold streamflow droughts in mid-March to April, both SMRI-1 and SMRI-3 reach more negative values than their SPI equivalents, which indicates that lacking snowmelt contributed to the streamflow drought generation (see Fig.9 g,h). Lacking snowmelt as contributing factor is further confirmed by considering the larger and rather persistent negative anomalies in both SWE and snowmelt in the preceeding 1 to 3 months (see Fig.9 e,f).

(x) L424: "Larger CWDs during..." is this a general (climatological) statement? The term "seasonal climatology" and the "(not shown)" addition might indicate this, but I recommend to stress this more clearly.

We rephrased the sentence to clarify the reference to climatology:

Based on the seasonal climatology of both temperature and evaporation (highest values during summer), larger absolute CWDs are generally expected to occur during the warm season (not shown).

(x) Starting sentence of Section 7: something is not quite right here with the references/model names.

We adjusted the references accordingly and changed the LaTeX-commands from *citet* to *citep*.

**Technical/editorial comments:**
We will implement all editorial suggestions.

() L9 onwards: inconsistent in naming all MeteoSwiss/SLF parameters, but only ERA5-Land as a whole.
(x) L12, "information on precipitation, evaporation-driven and streamflow deficits", can the authors please re-phrase; something seems not quite right here.

We rephrased this sentence as follows:

Deficits related to precipitation, evaporation, and streamflow are quantified using both standardized and non-standardized (drought/deficit) indices.

() L93/94: "n=18/94" I think it is misleading as the n refers to streamflow regime types first and then to individual catchments; why not simply include the numbers in prose? We will adapt this point

(x) L203, "variables" should go after "flux"?

We adjusted the section to be more precise in terms of variable distinction and conventions used:

For this, instantaneous variables and variables representing accumulations or fluxes are distinguished. For instantaneous variables, we provide daily average values. For variables representing accumulations and fluxes, we provide daily sums. Flux variables (mainly precipitation and evapotranspiration) are aggregated using the same temporal convention as RhiresD precipitation sums, i.e., from 06 UTC (day) to 06 UTC (day + 1) (see MeteoSwiss, 2021a). Instantaneous variables and ERA5-Land temperature were averaged from 00 UTC to 00 UTC again following the convention used in equivalent MeteoSwiss products (e.g., TabsD; MeteoSwiss, 2021b).

(x) L264: suggest to unify reference to R packages across the manuscript. E.g., elsewhere it is "cwd R-package", here it is "SCI-package".

We unified all package mentions to *packagename R-package*.

(x) L284: There is a left-over bracket, probably to be closed after "3.2" in L285. done
(x) L293: I suggest to spell out the z score more clearly instead of the "(value-mu)/sigma terminology.

We reformulated the sentence as follows:

Using the moving window climatology, standardized anomalies have been derived by first subtracting the climatological mean ($\mu$) and then dividing by the climatological standard deviation ($\sigma$) (also known as z-scores: $\frac{value-\mu}{\sigma}$.

(x) L308: "event definition", drop "definition"?

We changed the "event definition" to "variant".

For each variant, the event time series consists of consecutively numbered event phases and information on the event duration since the start.

() L362: MAM = march april may as elsewhere? In general, it is not clear what the acronym(s) is/are supposed to say.
(x) L383 An other > Another changed
(x) L424: "actual" in the sense of "non-anomaly" or in the sense of "non-potential"? Same for L431.

We slightly reformulated both sentences and explanations and also added subfigure references for clarity:

**L424:**
Similar to the longest streamflow drought phases, also the largest cumulative deficits in water balance (CWD) occurred between May–August 2022
**L431f:**
Cumulative deficits in potential water balance (PCWDs ) are more similar to cumulative streamflow deficits for the variable-threshold definition (CQD) . This reflects the different nature of CWDs and PCWDs. Deficits based on the actual water balance (P-E) are more strongly tied to the actual water availability and hence the individual streamflow (drought) phases.

(x) L563 and elsewhere: data set or dataset? changed (x) L563: "alsop" typo changed

**2    Comments reviewer 2**

This manuscript presents a comprehensive hydrometeorological dataset that is developed from a range of other, existing, datasets from various sources. This dataset is developed specifically to support assessment of drought and low flow conditions across catchments in Switzerland, and includes time series of a wide range of essential climate variables as well as derived drought indicators (e.g. SPI, SPEI, etc). As this dataset, or rather what I would consider a data collection can support detailed assessment of drought and the drivers of drought (see also use cases presented in the paper), I would think this in line with the scope of the journal.

Overall, the paper is well structured (with comments, see below) and the datasets that have been developed are outlined clearly, including the original data sources and general comments on data quality.

() One of the main concerns I have is that the dataset combines various underlying sources, in particular observational datasets and re-analysis datasets and datasets from models. Some attention is given at the start of the discussion on where care should be taken, but this is only discussed for selected variables, particularly related to snow (e.g. SWE), but much less discussed for other variables, for example E and PET (see also comments below). In this sense the discussion is somewhat poorly developed. The first part addresses some of the limitations, but a broader reflection on the quality of the dataset, including derived variables, would add to the depth of assessment of what is presented.
This is an important point and we will extend the discussion accordingly.

() Perhaps some indicative confidence level on the different datasets and combinations would be useful. I think this should also be added to the metadata provided with the dataset in Zenodo. Thank you for this suggestion, we will introduce three variable classes (L1 = direct observations and slight processing (e.g., spatial averages, L2 = variables from published data sets and slight processing, L3 = derived variables, obtained from input variables from different data sources).

() The dataset has been developed specifically for Switzerland, and for application in the Swiss context. It would be interesting to comment on how applicable the methods used/presented here would be applicable in other contexts. Some reflection on how applicable this could be in other contexts/countries, what the requirement are of underlying datasets would be etc. For example, use is made of ERA5-Land, which is of course available globally, but on the other hand the availability of observational data in CH is excellent. In other settings where there is less observational data, more use would then perhaps be made of re-analysis data. However, would this still make "sense"? This is indeed an important point and we will extend the discussion accordingly.

() The use cases that are presented in the paper are interesting and indeed demonstrate the utility of the dataset. On the other hand, they may not be the core of the paper, and the paper is very long One could consider including these in the supplementary material. What could then be interesting is to provide some general reflection on the application of these use cases, and how the HYD-RESPONSES dataset and methods have enabled these analyses, and why that was difficult or not possible prior to the development of this dataset.
We will move the use cases and their detailed discussion to the appendix.

**Detailed comments:**
() Line 98-99: Catchments are described as being at a certain elevation. This seems to be the mean elevation (this is mentioned later). May be useful for comprehension to name that here.

We added the following clarifications and additional information in Section 2:

**iii):**
In terms of mean catchment height, the catchments are distributed relatively equally between 500 and 2500 m a.s.l. with fewer (77 out of 98) catchments at elevation ranges above 1500 m a.s.l. Only eight catchments are higher than 2500 m a.s.l. and only one catchment is at very low elevation (catchment Wiese, Basel).

(x) Line 116: It may be useful to explain a bit better what "meaningful" means. I understand this is obtained through personal communication, but it is somewhat vague. see below
(x) Line 118: Stations where Q is measure at a water level station. I find this somewhat confusing, as I would presume (and my experience working with FOEN would confirm) that most river stations measure water levels and derive the discharge through a rating curve. Perhaps something else is meant. Please clarify. see below
(x) Line 118: Mention is made of NADUF stations. I am not familiar with what these are. Please provide some explanation.

We rephrased the reasoning for station inclusion/exclusion and added clarifications on what meaningful means. And yes, the reviewer is absolutely correct that streamflow (Q) is derived from water-level measurements by P-Q relationships. We rephrased the misleading statement for this

exclusion criteria. Further, we spelled out what NAWA is (formerly NADUF).

Here, meaningful is equivalent to stations which provide reliable streamflow (Q) time series and are associated with a physical/natural catchment. Stations have therefore be excluded if they i) only provide water-level information (no Q, 3 stations), ii) are not part of the main streamflow measurement network (e.g., stations from other networks such as the National Surface Water Monitoring Programme (NAWA; BAFU, 2023, 4 stations), iii) secondary stations (11 stations), iv) stations with potential return streamflow (= negative Q values, 2 stations), v) Q measured at derivations (2 stations), vi) stations without watershed delineation (i.e., subterranean; 1 station) and vii) uncertainties in time series composition due to displacement and/or temporarily missing Q of contributing stations (4 stations).

(x) Line 121: list of stations included

A list of all included stations in the HYD-RESPONSES dataset is already provided in the Appendix Tables A2-A5. Or has the reviewer an other list in mind (list of exclusions)?

(x) Line 124: May be useful to mention what is meant by assembled – I assume that this is compiling the catchment average for these variables.
We changed the wording from "assembled" to "extracted". But as the detailed methodology is presented in Section 4 (Data processing) we did not replicate further methodological details in this section.

() Line 126: I appreciate that the authors use the original names/ids of data depending on the source (e.g. RhiresD, tp). Table 1A provides some explanation which is useful. Perhaps it would be useful to provide in that table something like the WMO standard naming conventions. Would also be clear if the authors mention this strategy in the text, as it may otherwise become somewhat confusing.
We will clarify this point

(x) Line 139: Mentions is made of the quantile mapping approach being used. It is not so clear to me what the reference is for this bias correction through quantile mapping. Please clarify.

We slightly restructured the part on SPASS and removed the direct mentioning of quantile mapping (which is part of SnowQM):

The preliminary version was produced in 2022 and provides modeled and bias-corrected daily SWE data for the period September 1961–September 2022. The underlying SnowQM model is presented in detail in Michel et al. (2023). SWE is derived based on the daily TabsD and RhiresD products (see above). The spatial extent is restricted to the Swiss territory.

(x) Line 167: I think the word "The" at the start of the sentence needs to be dropped as it is otherwise not clear which digital soil suitability maps are intended as these have not been introduced.

We added "of Switzerland" to the sentence to make clear that we refer specifically to the maps used in the HYD-RESPONSES dataset which are introduced with this sentence.

The digital soil suitability maps of Switzerland provide information on a set of different soil characteristics assessed on 25 different geological and geomorphological units which are further discriminated by different landscape elements depending on aspect, slope and bedrock.

() Line 192: Does mean height here imply the mean elevation? Please be consistent.
We will clarify.

(x) Line 205: Check the sentence starting with "For accumulation... ". It is somewhat confusing and may need to be rephrased or elaborated to be clear.

We rephrased the sentence as follows:

For variables representing accumulations and fluxes, we provide daily sums.

(x) Line 215-216: Limnographs are often mentioned. The word is correct but to my mind not in such common use. It is somewhat a Germanism to my mind. Perhaps use Water level sensor or something similar
We have changed the name to your suggestion from limnograph to water level sensor.

(x) Line 233: Mention is made of interpolation between the day before the end of September and the day after, when SWE is det to zero. Surely the amount of water that melts is the same whether this happens over one or two days – and still unrealistically high, given that when resetting I assume all snow is considered as being melted. Perhaps I am misunderstanding the concept.

We clarified the paragraph as follows:

Note that SWE is reset in the SPASS dataset at the end of every snow year (every September $1^{st}$) to avoid unrealistically high accumulation of snow water equivalents ("snow towers") (see Michel et al., 2023). As snowfall and snowmelt were derived from daily differences in SWE ($\Delta$SWE), this reset can result in an artificial large negative $\Delta$SWE value on September $1^{st}$ that does not represent actual physical snowmelt. To prevent this model artifact from affecting the derived snowmelt time series, $\Delta$SWE values on September $1^{st}$ were set to missing values and replaced by linear interpolation using the $\Delta$SWE values from the preceding and following days.

() Line 239: Also related to the general comment. Here E and PET are derived from observed P (interpolated) and E and PET calculated in ERA5-Land. I am not clear how biases are dealt with, especially in E. If in ERA5-Land the precipitation is strongly biased, then surely E will (climatologically) tend to be too low in catchments that are water limited.
We will add a short discussion.

(x) Line 246: Here it seems to be suggested that PCWD is calculated using ERA5-Variables. Is that both for P and PET? Please clarify.

Yes, the HYD-RESPONSES dataset also provides PCWDs based on ERA5-LAND variables only (i.e., tp−pev). We have specified this in L246 as follows:

In some cases (especially for P–PET based only on ERA5-Land variables, i.e., tp−pev), PCWDs are not compensated each year and can persist over multiple years.

(x) Line 255: The word period is used here to indicate the accumulation window for SPI and SPEI. In other sections the word period is used to denote a period in time (e.g. 10 years). Please use

words consistently with a defined meaning, as it is otherwise somewhat confusing.

We changed the wording for standardized indices to aggregation (time) windows and to events where applicable.

(x) Line 273: mention is made of fewer than five DOYs flagged. I am not sure how these are flagged! Perhaps I missed it.

We clarified how the flags are obtained and what they signify by adding some additional information in the section:

The suitability of candidate distributions was assessed based on three indicators: the Shapiro-Wilks normality tests ($p$-values; Shapiro and Wilk, 1965), the number of flags returned by the fitting function *fitSCI* (see SCI R-package; Gudmundsson and Stagge, 2016), and the number of missing and/or implausible values. Implausible values are defined as values above or below $+3$ $(-3)$ STD following (Stagge et al., 2015). The returned flags in distribution parameter fitting were mainly related to convergence issues (non-convergence) (flag 3, see SCI R-package Gudmundsson and Stagge, 2016). Without a valid fit, the transformation to standardized index values is not possible resulting in missing values on the flagged DOYs in all time series years.

(x) Line 282: I was curious in the derivation of the distributional parameters of the distributions applied in SPI, SPEI and SMRI, if the same period of data was used to derive the parameters of the distribution, or if for each case the whole available time series was used. That could make comparison more difficult.

Yes, the same period of data was used to derive the parameters of the various distributions. We added additional clarifications to the section:

The distributions were fitted for each day of the year (DOY) based on the reference period 1991–2020. This results in a fit for each DOY derived from the same (window of) values for each distribution. Monthly SPI fits (SPI-1) are for example based on the 30 daily values up to the specific DOY for each of the 30 years in the reference period 1991–2020.

(x) Line 292: Mai à May
Adjusted.

(x) Line 336: Length – I guess of the main drainage path -please clarify.

we specified what length means as follows:

The first variant is derived based on a ratio between area and basin length $(A/L^2)$ known as form factor and the second variant is based on a ratio between the catchment area and the area of the circle with the smallest radius encircling the entire catchment $(A_{catch}/A_{circle})$ known as circularity ratio.

(x) Line 421: What does HRSg mean – often used but not clarified.

Hrsg. is the german equivalent for editors (Eds.). Only used for publications in german language. We adjusted Hrsg. to Eds. also for german sources.

(x) Line 441: Why the 15th percentile – if this is just to illustrate the please state is an arbitrary threshold.

We added the motivation for the choice of that percentile in the sentence:

Here, we present a composite analysis of median SPI values associated with streamflow droughts defined by the monthly $15^{th}$-percentiles of the streamflow which corresponds to the highest of the low-flow percentile used for the Swiss national drought platform .

() Line 490: It may be good to note that ERA5-Reanalysis and ERA5-Land are (to the best of my knowledge) not independent, with ERA5-Land derived from the former by downscaling using features such as elevation etc.
We will add a brief discussion.

() Line 499: The discussion that ends here is relevant, as in the dataset several indicators are developed that combine data from different sources - such as the cumulative water deficit, and the snowmelt corrected precipitation datasets. Given that these combine observational and reanalysis data, this may result in different levels of reliability of the derived datasets. I would be curious as to how is this flagged in these derived datasets. In other words, is some flag of degree of confidence set in the meta-data?
To be discussed.

(x) Table 3: I am not sure if Zenodo can be considered a provider – is this not more a repository?
We will change the wording where applicable.

(x) Line 563: also
Changed.

(x) Figure 4: "complete" is mentioned – I guess this is the same as full. Nice to be consistent.
Changed.

() Figure 5: The label NAs/Implausible should be described as to what it means (one can guess of course – but best to be clear).
We will clarify

**References**

[revised manuscript text omitted]